# Evaluating the potential for Haloarchaea to serve as ice nucleating particles

Jessie M. Creamean[1], Julio E. Ceniceros[3], Lilyanna Newman[2], Allyson D. Pace[2], Thomas C.J. Hill[1], Paul J. DeMott[1], and Matthew E. Rhodes[2]

[1]Dept. of Atmospheric Science, Colorado State University, Fort Collins, CO, USA
[2]Dept. of Biology, College of Charleston, SC, USA
[3]Geological Sciences, University of Texas, El Paso, TX, USA

*Correspondence to*: Jessie M. Creamean (jessie.creamean@colostate.edu) and Matthew E. Rhodes (rhodesme@cofc.edu)

**Abstract.** Aerosols play a crucial role in cloud formation. Biologically-derived materials from bacteria, fungi, pollen, lichen, viruses, algae, and diatoms can serve as ice nucleating particles (INPs), some of which initiate glaciation in clouds at relatively warm freezing temperatures. However, determining the magnitude of the interactions between clouds and biologically-derived INPs remains a significant challenge due to the diversity and complexity of bioaerosols, and limited observations of such aerosols to facilitate cloud ice formation. Additionally, microorganisms from the domain Archaea have to date not been evaluated as INPs. Here, we present the first results reporting the ice nucleation activity of four species in the class Haloarchaea. Intact cells of *Halococcus morrhuae* and *Haloferax sulfurifontis* demonstrated the ability to induce immersion freezing at temperatures up to –18 ˚C, while lysed cells of *Haloquadratum walsbyi* and *Natronomonas pharaonis* were unable to serve as immersion INPs. Exposure to heat and peroxide digestion indicated that the INPs of intact cells were driven by organic (*H. morrhuae* and *H. sulfurifontis*) and possibly also heat-labile materials (*H. sulfurifontis* only). While halophiles are prominent in hypersaline environments such as the Great Salt Lake and the Dead Sea, other members of the Archaea, such as methanogens and thermophiles, are prevalent in anoxic systems in seawater, sea ice, marine sediments, glacial ice, permafrost, and other cold niches. Archaeal extremophiles are both diverse and highly abundant. Thus, it is important to assess their ability to serve as INPs as it may lead to an improved understanding of biological impacts on clouds.

## 1 Introduction

Through their impact on the atmospheric energy budget and hydrological cycle, clouds play a prominent role in shaping Earth's climate at both global and regional scales (Baker and Peter, 2008; Boucher et al., 2013). However, due to the complexity of the microphysical processes associated with cloud formation and dynamics, clouds remain some of the most poorly constrained atmospheric features in climate models. In turn, while atmospheric aerosols strongly impact cloud formation, albedo, lifetime, and precipitation formation processes, disentangling the relationships and feedbacks among aerosols, clouds, and precipitation remains a significant challenge (Sato and Suzuki, 2019; Stevens and Feingold, 2009). In particular, the role of aerosols called ice nucleating particles (INPs), that induce glaciation in mixed-phase and ice clouds, is highly uncertain relative to other aerosol-cloud processes (Kanji et al., 2017). Improving understanding of such ice formation processes is crucial given most precipitation, globally, is initiated via the ice phase (Lohmann and Feichter, 2005).

In the Earth's troposphere, pure water remains in a supercooled liquid state until below –38 ˚C. At temperatures greater than –38 ˚C, the assistance of INPs such as mineral dust, volcanic ash, and select biologically-produced macromolecules (e.g., from pollen, fungi, bacteria, and other sources) is required to initiate the heterogeneous formation of primary ice embryos, that continue to grow

into larger ice crystals (Hoose and Möhler, 2012; Kanji et al., 2017). Depending on their surface properties and structural makeup, mineral dust and volcanic ash can raise the freezing point of water from –38 ˚C to as high as –12 to –15 ˚C (Murray et al., 2012). At temperatures above –15 ˚C, and specifically in the immersion freezing regime (i.e., heterogeneous glaciation of a supercooled

droplet), the majority of naturally-occurring INPs are biological in origin (Fröhlich-Nowoisky et al., 2016; Hoose and Möhler, 2012; Kanji et al., 2017; Morris et al., 2004). Intact and lysed cell wall fragments (Anderson and Ashworth, 1986; Du et al., 2017; O'Sullivan et al., 2015; Šantl-Temkiv et al., 2015), viable and nonviable cells (Anderson and Ashworth, 1986), and cellular byproduct materials such as exopolymeric substances, saccharides, and biosurfactants (Albers et al., 2017; Decho and Gutierrez, 2017; Demott et al., 2018; Dreischmeier et al., 2017; Perkins et al., 2020b; Zeppenfeld et al., 2019) have all been demonstrated to

serve as biologically-derived INPs (Després et al., 2012; Fröhlich-Nowoisky et al., 2016). The most thoroughly-studied biologically-derived INPs of measurable abundance are strains of the bacterium *Pseudomonas syringae*, which is capable of forming ice at temperatures as high as –1 ˚C (Maki et al., 1974; Morris et al., 2008). The ability of *P. syringae* to effectively facilitate ice formation is due to a specific ice nucleating protein (i.e., a protein that is ice nucleation active, designated Ina), that coordinates the crystallization of ice (Cochet and Widehem, 2000; Davies, 2014; Failor et al., 2017; Gurian-Sherman and Lindow,

1993; Hartmann et al., 2013; Šantl-Temkiv et al., 2015). The Ina protein is found in select lineages of Bacteria all within the class Gammaproteobacteria (Warren, 1995). There is some evidence that slightly less effective, but highly abundant proteinaceous ice nucleating proteins can be found within fungi, but they remain to be fully categorized (Fröhlich-Nowoisky et al., 2015; Kunert et al., 2019). Further, a recent study identifying the first Gram positive ice nucleating bacterium from cloud precipitation suggests that ice nucleating proteins are far more widespread within the bacterial domain than previously thought (Failor et al., 2017).

Even though many laboratory and field-based investigations have alluded to the importance of biologically-derived INPs, the relatively limited available observational data have caused models to produce equivocal results regarding the global significance of biological ice nucleation in cloud and precipitation formation (Burrows et al., 2013; Hoose et al., 2010b; Hummel et al., 2018; Phillips et al., 2009; Sesartic et al., 2012; Twohy et al., 2016; Vergara-Temprado et al., 2017). This modelling issue is further complicated by a very limited understanding and representation of secondary ice formation processes and their links to biologically-

derived INPs in clouds. Climate models of all scales require information on INP sources to accurately represent ice nucleation and thus cloud microphysics, especially considering: (1) biologically-derived INPs form ice at cloud temperatures as high as –1 ˚C while certain mineral dusts glaciate modestly starting at –12 ˚C and (2) biologically-derived and mineral INP concentrations can vary by several orders of magnitude at any given temperature (Burrows et al., 2013; Demott et al., 2016; Hoose et al., 2010a; Hoose et al., 2010c; Kanji et al., 2017; Mccluskey et al., 2019; Petters and Wright, 2015; Vergara-Temprado et al., 2017).

While there have been ongoing efforts to identify and characterize INPs in the eukaryotic and bacterial domains (Dreischmeier et al., 2017; Failor et al., 2017; Fröhlich-Nowoisky et al., 2016; Hill et al., 2014; Kanji et al., 2017; Kunert et al., 2019; Ling et al., 2018; Morris et al., 2008; Pummer et al., 2012; Qiu et al., 2019), to date, there has been no study that attempts to identify INPs within the domain Archaea. This is in no small part due the fact that, until recently, the Archaea were believed to be largely relegated to marginal existences in extreme environments on our planet. Though recent studies have begun to reveal the true

ubiquity and abundance of Archaea in Earth's ecosystems—including seawater, ocean sediment, plankton, soil, marine and terrestrial biofilms, and sea ice, where they can comprise up to 40% of the microbial taxa in an ecosystem (Cavicchioli, 2011; Flemming and Wuertz, 2019; Hoshino and Inagaki, 2019; Junge et al., 2004; Mondav et al., 2014; Munson et al., 1997; Ochsenreiter et al., 2002; Santoro et al., 2019)—many Archaea remain uncultured and unculturable in laboratory settings, further complicating investigations into their possible propensities to serve as INPs. At the same time, however, the archaeal domain

contains both unique cell wall compositions and cell surface structures not present in the other domains (Albers and Meyer, 2011).

While appearing highly similar to the bacterial domain in both size and appearance, the archaeal cell envelope is distinct in several ways. In contrast to the Bacteria, most cultured Archaea maintain a proteinaceous surface layer, or S-layer, that provides the cell with structural stability. In many Archaea, the S-layers provide the entirety of the cell envelope. Comparatively few Archaea contain additional cell envelope polymers, and the ones that do, do not produce the near ubiquitous bacterial polymer peptidoglycan. Some Archaea do produce a structurally similar polymer, pseudomurein. The archaeal S-layers are often composed of a single protein or glycoprotein arranged in symmetrical patterns, with hexagonal symmetry being the most common (Albers and Meyer, 2011). As with the Bacteria, many surface exposed proteins are modified in a variety of ways. In addition to the S-layer itself, the archaeal domain contains its own regimen of surface proteins and structures that interact with the external environment. Thus, an entire domain worth of cell surface structures remains to be assessed for potential INP activity.

Here we present a first attempt to assess the potential for members of the domain Archaea to serve as INPs. We initiated our investigation with four members of the obligate halophilic lineage, the Haloarchaea: *Halococcus morrhuae, Haloferax sulfurifontis, Haloquadratum walsbyi,* and *Natronomonas pharaonis*. Together, these four Haloarchaea represented a variety of cell surface designs. Halococcus is one of a limited number of genera belonging to the Archaea that does not possess an S-layer. Instead they possess a cell envelope that includes highly sulphated heteropolysaccharides (Schleifer et al., 1982). They can be several microns in size, exhibit a cocci (i.e., spheroidal) morphology, and do not lyse in fresh water (Legat et al., 2010; Leuko et al., 2004). *Haloferax* possesses a exclusively sulphur-rich S-layer and exists as rod- or irregular-shaped cells several microns in size (Elshahed et al., 2004). *Haloquadratum* can produce halomucin in addition to its S-layer and grows in its trademark square morphology, a couple microns in size (Burns et al., 2007). *Natronomonas* exist as rods several microns in length and prefer alkaline hypersaline environments (Falb et al., 2005). Specifically, *H. walsbyi* and *N. pharaonis* are particularly sensitive to lysis under hyposaline conditions (Boring et al., 1963)—they readily lyse in salinities of roughly 5% salt by weight and below. Assessing a variety of cells that lyse or remain intact is relevant for ice nucleation because cell fragments of other microorganisms have been shown to serve as INPs and (Anderson and Ashworth, 1986; Du et al., 2017; O′Sullivan et al., 2015; Šantl-Temkiv et al., 2015) and archaea might lyse naturally once exposed to atmospheric water vapor in the aerosol phase.

We opted to initiate our investigation with members of the Haloarchaea for a variety of reasons, including: (1) the diversity of cell surface compositions, (2) they are relatively easy to culture compared to other archaeal lineages, and (3) the regional dominance of the Haloarchaea in relatively large hypersaline environments. While few largescale geographic areas are dominated by members of the domain Archaea, one notable exception is hypersaline bodies of water such as the north basin of the Great Salt Lake and the Dead Sea. These waterbodies extend over hundreds of square kilometres, often contain upwards of 90% Archaea, and can impact the local climate and weather patterns (Carpenter, 1993). Thus, when investigating the impact of potential archaeal INPs in an environment, halophiles offer attractive starting points.

To fully understand the interaction of Earth's climate with the microbial world, it is imperative to include the impact of the archaeal domain, since even when less prevalent, the possession of ice nucleation activity will enable a species to exert an outsized effect on its environs. And to fully understand the potential impact of the archaeal domain on Earth's climate, it is important to assess the potential for members of the Archaea to serve as INPs and contribute to cloud formation.

## 2 Materials and methods

### 2.1 Cell cultures

Cell cultures for all four haloarchaeal species tested were purchased from the DSMZ-German Collection of Microorganisms and Cell Cultures (https://www.dsmz.de/) and the associated saline medias were prepared for *H. morrhuae* (medium 97)*, H. sulfurifontis* (medium 1018)*, H. walsbyi* (medium 1091)*,* and *N. pharaonis* (medium 371) with the following alterations: (1) for medium 97 only 150 g of NaCl was added instead of 250 g NaCl for growth of *H. morrhuae* and (2) one gram per litre of glycerol was supplemented to medium 1091 for growth of *H. walsbyi.* The salinities of the media were confirmed using a handheld refractometer. All cultures were grown at 37 ˚C and 100 rpm until mid-log phase. The purity and cell density were monitored optically using a Leica DM750 microscope (https://us.leica-camera.com/) at 1000x magnification with a 100x oil immersion objective lens on a Petroff-Hausser 3900 counting chamber (http://hausserscientific.com/). Cells were counted and monitored for growth until mid-log phase at which point, they were shipped on ice overnight to Colorado and stored for up to 48 hours at 4 ˚C. Cultures were checked a final time for cell density immediately prior to ice nucleation assays to ensure that no appreciable growth had occurred during transport and storage. Table 1 provides the cell concentrations and salinities of all four prepared cultures immediately prior to ice nucleation assays.

### 2.2 Preparation of samples for ice nucleation measurements

Due to the high salinities of the media which would have caused significant freezing point depression—for reference, seawater is typically 30 – 35 ppt (Bodnar, 1993) and can depress the freezing point of water by $\geq 2$ °C (Irish et al., 2019; Schnell and Vali, 1975)—samples were diluted as shown in Table 1. Reductions in salinity can inherently cause the cells of certain haloarchaeal species to lyse (Boring et al., 1963; Legat et al., 2010; Leuko et al., 2004). Testing ice nucleation responses from cell lysis is relevant given: (1) cell fragments from fungi and bacteria have been previously observed to serve as INPs (Anderson and Ashworth, 1986; Du et al., 2017; O′Sullivan et al., 2015; Šantl-Temkiv et al., 2015) and (2) a less saline environment is more atmospherically relevant for how archaea might behave once exposed to atmospheric water vapor in the aerosol phase. Cell lysis was determined by diluting cultures with deionized water as shown in Table 1 and microscopically assessing cellular survival. Samples were diluted 1:15 with deionized water to result in 1.1 – 1.4% salinity. This dilution was chosen based on tests using saline solution controls (made with Instant Ocean® Sea Salt) and solutions with *H. walsbyi* and *N. pharaonis* whereby, for example, freezing would not be observed until $< -23$ °C and then only reach $\sim 0.2$ fraction frozen at the instrument lower limit for a 10% salt solution. Thus, a 1:15 dilution was found to be a good compromise for preventing significant freezing point depression while still obtaining full spectra (i.e., reaching a fraction frozen of 1). Additional tests were conducted on *H. morrhuae* cultures diluted to 1:6 and 1:30 with cells remaining intact at all dilutions tested. Cultures were checked microscopically after dilution for cell density and to ascertain if cell lysis had occurred.

### 2.3 Immersion freezing ice nucleation experiments

INP concentrations were measured using the Colorado State University (CSU) Ice Spectrometer (IS) (Hiranuma et al., 2015; Mccluskey et al., 2018; Suski et al., 2018), which is an immersion freezing measurement device suited to testing aliquots of liquid culture. The IS is constructed using two 96-well aluminium incubation blocks, designed for incubating polymerase chain reaction (PCR) plates, placed end-to-end and encased on their sides and base by cold plates. Immersion freezing temperature spectra were obtained by dispensing 50-µL aliquots of cell suspensions into sterile, 96-well PCR trays in a laminar flow cabinet. Each sample consisted of 24 aliquots. PCR plates were then placed into the blocks of the IS, after which the device was cooled using Sytherm XLT in a recirculating low temperature bath. Frozen aliquots were detected with a CCD camera system controlled with LabVIEW

(LabVIEW; NI, Inc.) as temperature was lowered at approximately 0.33 °C min$^{-1}$ to down to –29 °C. The temperature uncertainty of the IS is less than ±0.2 C, which is a combination of the uncertainty in the thermocouples and the temperature variation across the blocks due to gradients in cooling. From the number of wells frozen at each temperature step, the fraction frozen is calculated. Confidence intervals (95%) were calculated based on the methodology of Agresti and Coull (1998). Deionized water and uninoculated diluted media were run as controls for all species.

Heat and peroxide treatments were conducted to isolate heat-labile (e.g., proteinaceous) and organic INPs in the diluted samples (Barry et al., 2021; Creamean et al., 2020; Hill et al., 2016; Mccluskey et al., 2018; Perkins et al., 2020a; Suski et al., 2018; Tobo et al., 2014). Each sample (i.e., all dilutions listed in Table 1) was subject to heat and peroxide treatments to obtain the heat-labile (proteinaceous) and organic frozen fractions in addition to the total (unamended) frozen fractions. The stability (or lack thereof) of INPs to these treatments provides an indication of composition. To assess the contribution of heat-labile entities, a 1.5-mL aliquot of suspension was tested after heating to 95 °C for 20 min. To remove all organic INPs, 0.75 mL of 30% $H_2O_2$ was added to another 1.5-mL aliquot of suspension and the mixture heated to 95 °C for 20 min while illuminated with UVA/UVB fluorescent bulbs (Exo Terra Reptile UVB, 2 × 26 W providing ~2,000 µW cm$^{-2}$ UVA and ~300 µW cm$^{-2}$ UVB at the distance used) to generate hydroxyl radicals (residual $H_2O_2$ was removed using catalase), prior to testing. Both heat and peroxide treated samples were tested simultaneous to the unamended samples in the IS for each species, since the CSU IS can house up to four 96-well plates at one time. Remaining INPs are possibly aggregates of cellular material that are not fully digested, inorganic INPs in the media, or other biological materials resistant to heat and peroxide digestion (Conen et al., 2011; Perkins et al., 2020a).

## 3 Results and discussion

### 3.1 Interspecies comparison of haloarchaeal ice nucleation abilities

Figure 1 shows the results of the 1:15 dilutions of each of the haloarchaeal species. The media for all four species contributed modestly to background freezing as compared to deionized water controls. *H. morrhuae* performed best as an INP, initiating the freezing of water at temperatures as high as –17.6 ˚C. *H. sulfurifontis* also demonstrated enhanced capability to serve as an INP, freezing water at temperatures above that of deionized water and sterile media, with a freezing onset of –19.2 ˚C. These haloarchaea are not as proficient ice nucleators as other more commonly-studied biologically-derived entities such as certain bacteria (up to –1.3 ˚C) (Kim et al., 1987; Lindow et al., 1989; Maki et al., 1974; Vali et al., 1976), fungi (up to –1 ˚C) (Kunert et al., 2019; Richard et al., 1996), lichen (up to –2 ˚C) (Kieft, 1988; Kieft and Ruscetti, 1990), and pollen (up to –8 to –12 ˚C) (Hader et al., 2014; Pummer et al., 2011). The haloarchaeal species fall closer in line with less effective biologically-derived INPs such as fungal spores (onset freezing temperatures reported up to –10 ˚C, but typically initiate freezing < –25 ˚C) (Iannone et al., 2011; Jayaweera and Flanagan, 1982) and diatoms (observed up to –24 ˚C) (Knopf et al., 2011). *H. morrhuae* cells were also more effective at nucleating ice even though total cell concentrations were three times higher for *H. sulfurifontis* (Table 1). One possible explanation is that *H. morrhuae* is unusual among the Archaea in that it has a cell envelop composed of polysaccharides, which have been shown to serve as tracers for ice nucleating activity (Zeppenfeld et al., 2019). In general, although not "warm temperature" INPs (i.e., that glaciate > –15 ˚C), these haloarchaea are comparable in ice nucleation activity to fungal spores and diatoms and can thus contribute to INP populations at very relevant atmospheric freezing temperatures.

As expected, ice nucleation activity did not occur in all tested haloarchaeal species, just as with bacteria (Karimi et al., 2020; Szyrmer and Zawadzki, 1997). Lysed cells of both *H. walsbyi* and *N. pharaonis* did not exhibit ice nucleation activity as the fractions frozen were not higher than the media blanks. Interestingly, the dilution of the log-phase cultures to 1:15 with deionized

water resulted in the complete lysis of *H. walsbyi* and *N. pharaonis* as opposed to *H. morrhuae* and *H. sulfurifontis,* which both remained intact. Testing ice nucleation responses from cell lysis is relevant given: (1) cell fragments from fungi and bacteria have been previously observed to serve as INPs (Anderson and Ashworth, 1986; Du et al., 2017; O'Sullivan et al., 2015; Šantl-Temkiv et al., 2015) and (2) a less saline environment is more atmospherically relevant for how haloarchaea might behave once aerosolized and exposed to atmospheric water. However, the results presented here indicate that of the species studied, lysed haloarchaeal cells (i.e., cell fragments) do not enhance ice nucleation abilities, and possibly even suppress it. This is analogous to previous work on bacteria, whereby it is well known that lysing of ice nucleation active bacterial cells decreases the efficiency at which they are INPs (e.g., Lindow et al., 1989).

**3.2 Response of haloarchaeal species to heat and peroxide treatments**

Both *H. morrhuae* and *H. sulfurifontis* demonstrated responses to both the heat and peroxide treatments. Figure 2 shows the spectra for the 1:6, 1:15, and 1:30 samples (i.e., increasing dilutions or decreasing cell densities) of *H. morrhuae*. The ice nucleating activity of all three samples was reduced by heat by 0.3 – 0.9 ˚C on average (i.e., when calculating the average freezing temperature per spectrum and subtracting from average freezing temperature of the unamended INP spectrum, per sample), but especially reduced by peroxide (i.e., by 1.5 – 4.2 ˚C on average). These differences are statistically significant for decreases in fraction frozen of 0.25 or more at similar temperatures when applying Fisher's Exact test ($p < 0.0479$), which was the case for all data except for the first three and last data points for the 1:15 *H. morrhuae* unamended and heat spectra (Figure 2b; Table S1). These results indicate the samples contained some heat-labile, likely proteinaceous, INPs, but contained a relatively larger contribution from other biogenic organic INPs (Conen et al., 2011; Hill et al., 2016; Mccluskey et al., 2018). Interestingly, increasing the dilution, such that the sample contained lower cell densities, led to a larger decrease in fraction frozen from the peroxide treatments. For the 1:6 sample (i.e., highest cell density), frozen fractions dropped 0.9 ˚C and 1.5 ˚C on average for the heat and peroxide treatments, respectively. For the largest decrease, the 1:30 sample (i.e., lowest cell density) dropped by 0.7 ˚C and 4.2 ˚C on average for heat and peroxide treatments, respectively. One conceivable explanation for the increased efficacy with decreasing cell density is that the peroxide—the same volume was used in each sample, based on successful peroxide sample degradation reported in previous studies (e.g., Barry et al., 2021; Creamean et al., 2020; Suski et al., 2018)—remained in higher concentration to digest a lower concentration of cells; thus, less residual organic material was available to serve as INPs. It is possible that higher volumes of peroxide would be needed for higher cell concentrations to eliminate all organic material, depending on the location of the organic material (i.e., if it is extra- or inter-cellular). A recent study demonstrated excess peroxide was required to effectively reduce INP concentrations to background levels for the lignin biopolymer (Bogler and Borduas-Dedekind, 2020). We recommend future work evaluating archaeal INPs should involve a more rigorous peroxide treatment regimen to test this hypothesis.

For the other three haloarchaeal species, only *H. sulfurifontis* exhibited a decrease in frozen fractions—fraction frozen dropped by 2.7 ˚C and 1.8 ˚C on average for heat and peroxide, respectively (Figure 3). *H. walsbyi* and *N. pharaonis* showed no response to the treatments because there were essentially no INPs to begin with, thus these species are not discussed herein. For comparison between the *H. sulfurifontis* unamended and heat spectra, data > –22 °C and < –25 °C are the only data not statistically significant. For comparison between the unamended and peroxide spectra, only data < –26 °C are not statistically significant. These decreases were mostly observed and statistically significant when the fraction frozen was approximately 0.2 to 0.8 (average decrease in freezing temperatures within this range was 3.8 ˚C and 2.3 ˚C, respectively), as opposed to *H. morrhuae*, where the difference in temperature from the unamended to the treated frozen fractions were roughly equivalent throughout the spectra. Interestingly, *H. sulfurifontis* spectra were more responsive to the heat treatment than to peroxide (i.e., exhibited a larger decrease in average

freezing temperatures for heat than for peroxide), indicating this species contained more heat-labile, probably proteinaceous, INPs versus organic INPs—the opposite of *H. morrhuae*. Collectively, these results indicate that *H. morrhuae* contained more organic relative to heat-labile INPs, while *H. sulfurifontis* contained more heat-labile as opposed to organic INPs. These haloarchaea have very different cellular envelop compositions: *H. sulfurifontis* contains a proteinaceous S-layer while *H. morrhuae* is devoid of such an S-layer but instead possessed a cell envelope that is composed of highly sulphated heteropolysaccharides. Thus, it would make sense that *H. sulfurifontis* is more sensitive to heat than peroxide given its proteinaceous cell envelope (assuming those proteins are ice nucleation active) and *H. morrhuae* is more sensitive to peroxide than heat given its polysaccharide-rich cell envelope.

## 4 Conclusions

Here, we present the first reported results on the ice nucleation activity of the domain Archaea. Specifically, we focus on four species within the Haloarchaea due to their diversity of cell surface compositions, ease of culturability, and regional presence within relatively large hypersaline environments. The freezing temperature ranges measured for the haloarchaea species involved in this study are put into broader context by comparing to reported ranges for other known biologically-derived INPs (Figure 4). While not the most proficient of biologically-derived INPs such as lichen, bacteria, fungi, viruses, algae, pollen, or pollen wash water, which nucleate ice at warmer temperatures, haloarchaea fall within moderate freezing temperature ranges above phytoplankton exudates, diatoms, and fungal spores. However, this work is based on a limited subset of species and future work should focus on other members of the Archaea.

All haloarchaea species were introduced to hyposaline conditions to reduce freezing point depression. While intact cells of *H. morrhuae* and *H. sulfurifontis* demonstrated ability as INPs, inducing freezing up to –18 ˚C, lysed cells of *H. walsbyi* and *N. pharaonis* did not exhibit ice nucleation activity over the temperature range tested. The intact cells that demonstrated ice nucleation activity contained both heat-stable organic ice nucleating entities (*H. morrhuae*) and heat-labile material (*H. sulfurifontis*). This may be negated at cloud level salinities where *H. sulfurifontis* would be expected to lyse as well. *H. morrhuae* on the other hand remained intact at all salinities (Legat et al., 2010; Leuko et al., 2004). It is also rare among both the Haloarchaea, in particular, and the Archaea, as whole, in that its surface is completely devoid of a proteinaceous S-layer. Ice-binding ability is a characteristic of both ice nucleating and antifreeze proteins and is influenced primarily by their size (Eickhoff et al., 2019; Qiu et al., 2019). Therefore, further work is needed to directly evaluate the surface properties to both disentangle which constituents are responsible for ice formation in the *Halococcus* and assess the ice nucleating potential within other members of the Haloarchaea and Archaea, in general.

While halophilic archaea are prominent in hypersaline environments throughout the globe, such as the Great Salt Lake and the Dead Sea, other members of the domain Archaea such as methanogens and thermophiles are prevalent in anoxic systems in seawater, sea ice, marine sediments, glacial ice, permafrost, hot springs, submarine hydrothermal vents, and hot, dry deserts (Amend and Shock, 2001; Collins et al., 2010; Oremland and Taylor, 1978; Price, 2007; Thauer et al., 2008; Thummes et al., 2007; Van Der Maarel et al., 1999). However, some studies allude to the fact that these extremophiles are not confined to extreme living conditions, which qualifies them as one of the most abundant prokaryotes on Earth (Delong, 1998). Thus, these microorganisms are more ubiquitous than one might think, are present in the atmosphere (Fröhlich-Nowoisky et al., 2014), and may affect cloud formation (Amato et al., 2017). Indeed, the order Halobacteriales, which contains *H. morrhuae*, has been found to be present in continental air and was relatively abundant among the Archaea found in marine air (Fröhlich-Nowoisky et al., 2014). Further, Archaea accounted for several percent of all sequences in boundary layer air sampled from 45 – 50 °S over the Southern Ocean

(Uetake et al., 2020). Future work should focus on characterizing a wide range of environmentally-relevant Archaea, such as methanogens (see Creamean et al., 2020) and ammonia oxidizers (see Fröhlich-Nowoisky et al., 2014), for their ice nucleating properties and address how they may be important for regional cloud formation, and hence weather and climate, as the impacts of the Archaea on climate processes is currently unknown.

**Acknowledgments:** The authors would like to acknowledge Prof. Stephen Schmidt and Dr. Pacifica Sommers of the University of Colorado, Boulder for assisting with preservation and culturing of the haloarchaea after arrival in Colorado. J. Ceniceros was supported by the U.S. Department of Commerce, National Oceanic and Atmospheric Administration, Educational Partnership Program under Agreement No. NA16SEC4810006. This work was funded by a NASA EPSCoR Research Infrastructure Development for the state of South Carolina.

**Table 1. Samples produced from all four haloarchaeal species for ice nucleation testing. Initial sample cell concentrations and salinities are provided after culturing. Diluted cell concentrations and salinities were calculated based on the volume of culture mixed with media and dilution factor. The cell state is also provided after dilution.**

| Haloarchaeal species | Initial cell concentration (mL$^{-1}$) | Initial salinity in ppt (%) [**] | Dilution factor | Diluted cell concentration (mL$^{-1}$) | Diluted salinity in ppt (%) [**] | Cell state after dilution |
|---|---|---|---|---|---|---|
| *Halococcus morrhuae* | $2.7 \times 10^9$ [*] | 160 (16.0%) | 1:6 | $4.5 \times 10^8$ | 26.7 (2.7%) | intact |
| | | | 1:15 | $1.8 \times 10^8$ | 10.7 (1.1%) | intact |
| | | | 1:30 | $9.0 \times 10^7$ | 5.3 (0.5%) | intact |
| *Haloferax sulfurifontis* | $3.5 \times 10^9$ [*] | 177 (17.7%) | 1:15 | $5.8 \times 10^8$ | 11.8 (1.2%) | intact |
| *Haloquadratum walsbyi* | $2.4 \times 10^8$ | 210 (21.0%) | 1:15 | n/a [***] | 14.0 (1.4%) | lysed |
| *Natronomonas pharaonis* | $1.1 \times 10^9$ [*] | 207 (20.7%) | 1:15 | n/a [***] | 13.8 (1.4%) | lysed |

[*] Initial cell concentrations were too many to count optically and were estimated based on diluted samples, which were 1:15 in deionized water dilution for *H. morrhuae* and 1:6 for *H. sulfurifontis* and *N. pharaonis*.

[**] For initial and diluted salinity, "ppt" represents parts per thousand.

[***] For the diluted cell concentrations, "n/a" = not applicable due to lysing.

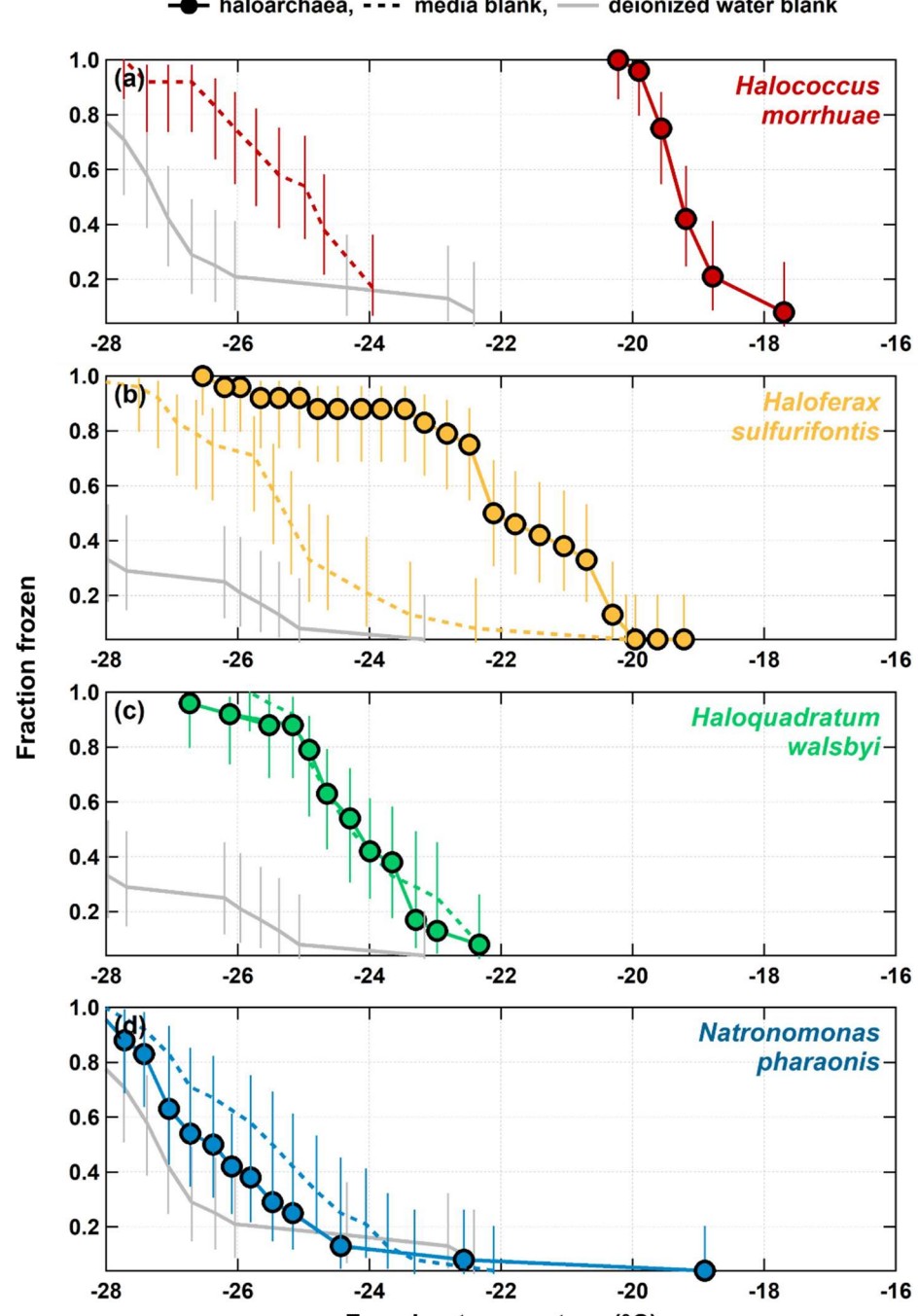

Figure 1. Freezing spectra for each of the haloarchaeal species diluted 1:15 in deionized water: (a) *H. morrhuae*, (b) *H. sulfurifontis*, (c) *H. walsbyi*, and (d) *N. pharaonis*. *H. morrhuae* and *H. sulfurifontis* did not lyse, *H. walsbyi* lysed, and *N. pharaonis* partially lysed. Note that *H. walsbyi* and *N. pharaonis* do not reach a frozen fraction of 1 because not all drops were frozen at the lower limit of the IS tests. Error bars indicate 95% confidence intervals.

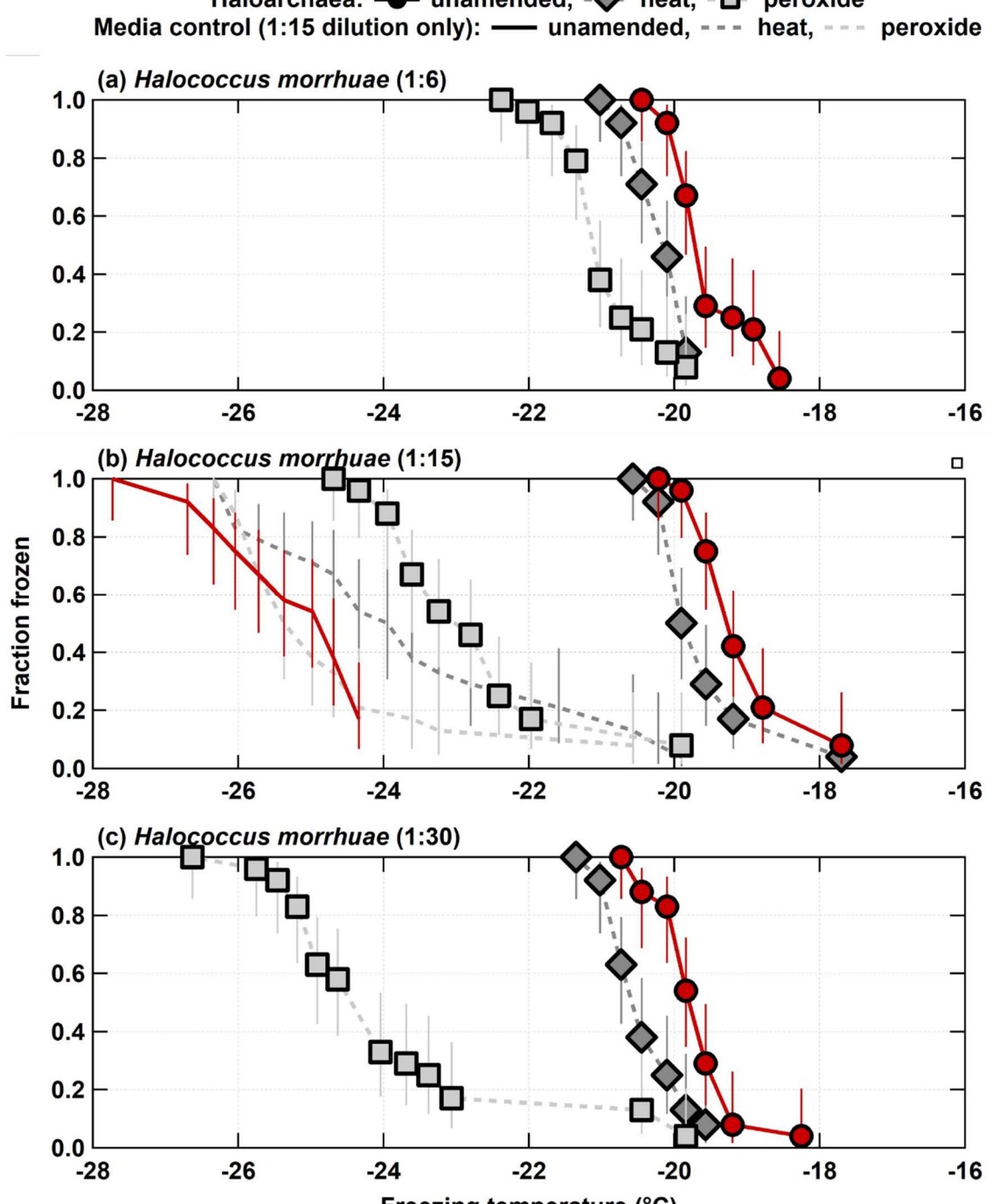

**Figure 2. Unamended, heat-labile (heat), and organic (peroxide) frozen fractions for each of the three *H. morrhuae* dilutions from the processing treatments. Note media controls were only conducted for the 1:15 dilution. Error bars indicate 95% confidence intervals.**

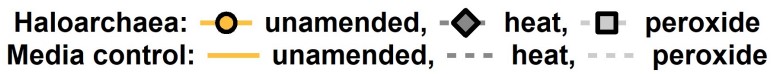

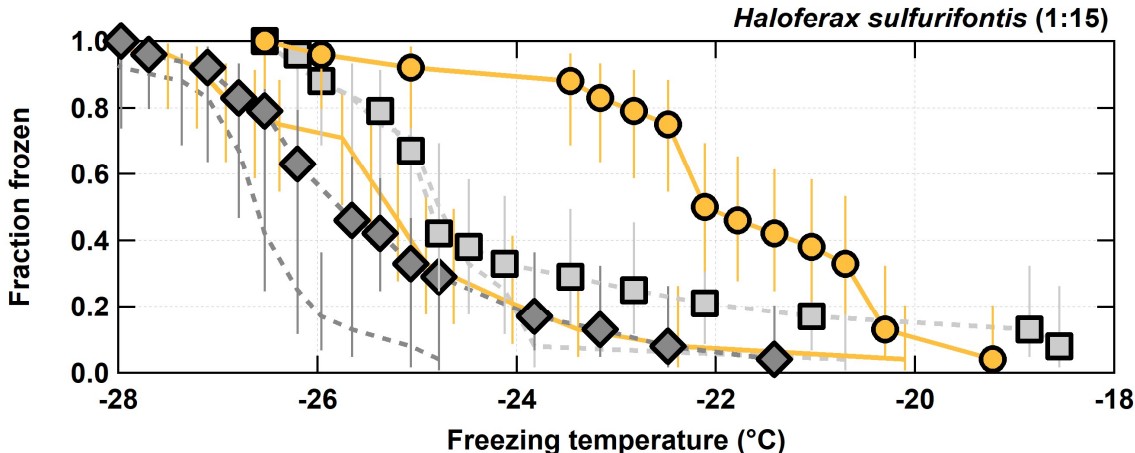

Figure 3. Unamended, heat-labile (heat), and organic (peroxide) frozen fractions for each of the 1:15-diluted *H. sulfurifontis* from the
processing treatments. *H. walsbyi* and *N. pharaonis* are not shown since they exhibited no INP activity in their unamended spectra. Error
bars indicate 95% confidence intervals.

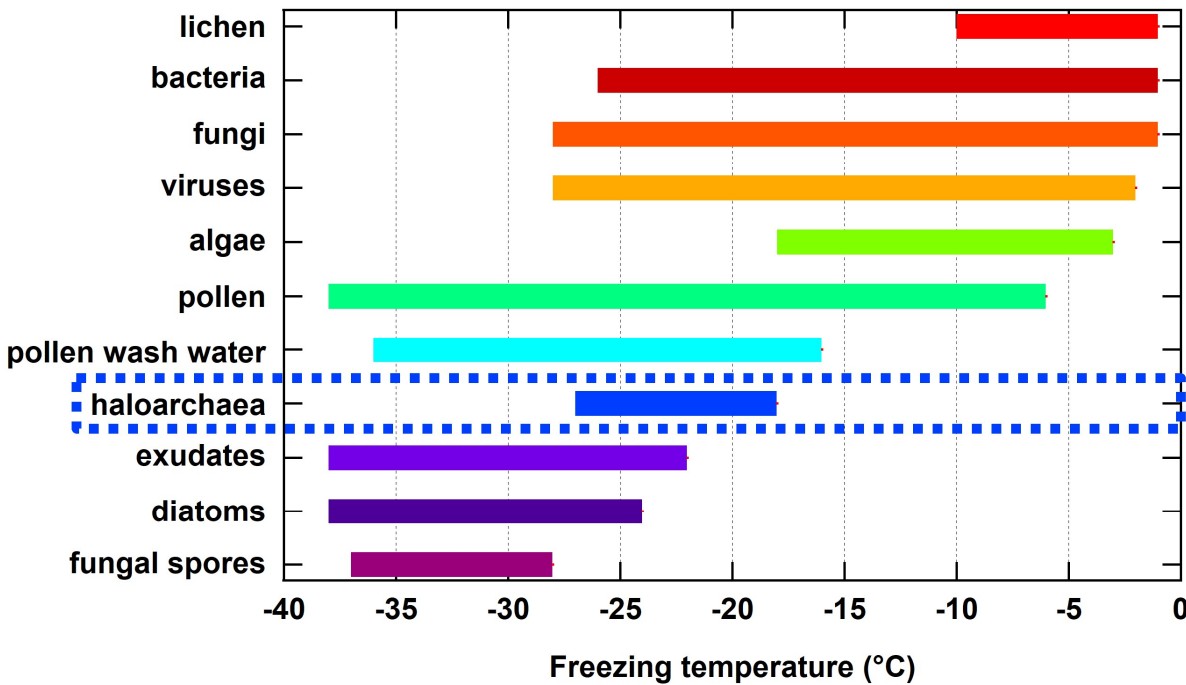

**Figure 4. Summary of approximate freezing temperature ranges reported for known biologically-derived INPs. Haloarchaea data are those from *H. morrhuae* and *H. sulfurifontis* from the current work. Data for lichen, bacteria, fungi, algae, pollen, pollen wash water, phytoplankton exudates, diatoms, and fungal spores were obtained and compiled from reviews by Després et al. (2012), Huang et al. (2021), Kanji et al. (2017), and Murray et al. (2012), and references therein. Data for viruses are more limited and were obtained from Adams et al. (2021) and Orser et al. (1985). Note that freezing temperature ranges for each result from one to multiple studies and may be dependent on the technique used (e.g., instrumental freezing limits and drop size).**

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
