# Peer review of "Evaluating the potential for Haloarchaea to serve as ice nucleating particles"

_Biogeosciences, 2020_

## Referee Comment (RC1) · Anonymous Referee #1 · 30 Nov 2020

General comments:

Creamean at al. studied ice nucleation activity within the domain Archaea and are the first to report ice nucleation activity of this domain. Up to date, Archaea have not been evaluated as INP. In two out of four investigated species, the ability to induce freezing above -18 °C was found. The authors performed additional experiments (heat treatment, peroxide digestions) to further study the composition of INPs from Archaea. They suggest that the IN activity of intact cells were driven by organic and heat-labile materials.

This work provides valuable insights into ice nucleation activity within the domain Archaea and contributes to the understanding of biological ice nucleation as a whole. The results are certainly publishable and are aligned with the scope of the journal.

Overall, the manuscript is well-written and -structured, and I recommend to publish it in Biogeosciences after the following comments have been addressed.

Specific comments:

Line 111: Where did you obtain the cell cultures from? Please provide more information.

Line 117: How did you determine the salinities?

Line 122, Table 1, Table 2: Usually ppt is understood as "parts per trillion". In specific disciplines such as oceanography, however, the abbreviation ppt is commonly used for "parts per thousand". I would recommend additional annotations to minimize misunderstandings.

Line 137: What is the temperature uncertainty of the measurement?

Line 138: If the freezing of wells was recorded in 0.5 °C steps, how did you receive initial freezing temperatures such as -17.6 °C or -19.2 °C (lines 152-153)? Why do you have often more than two data points within one degree in all figures?

Line 140: How many independent experiments have been performed? Have the results been checked for reproducibility? Are the data presented in the figures arithmetic mean values? Otherwise, how did you calculate the standard error?

Line 142: For simplification, I would recommend to use the terms "heat" and "peroxide" (treatment) throughout the whole manuscript as they are also used in the figure legends.

Line 172: Is there any proof that the intact cells of H. walsbyi and N. pharaonis would show ice nucleation ability, which can be suppressed by lysis? You could additionally test if the ice nucleation ability of H. morrhuae and H. sulfurifontis can be suppressed upon lysis.

Lines 177, 180-182, 187-188: Which temperature do you refer to? Initial freezing

temperature? T50?

Lines 182-186: Have you tried to use an excess of peroxide to remove all organic material, which could serve as INP, so that there are no differences between sample concentrations anymore?

Lines 187-188: Please refer to Figure 3 here.

Lines 190-191: Please clarify: "Interestingly, H. sulfurifontis INP spectra were more responsive to the heat treatment as opposed to peroxide, . . .". Assuming that the same symbols and gray colors were chosen for figures 2 and 3 (legend is missing), the peroxide treatment shows a stronger reduction of INP spectra than the heat treatment.

Line 197: Can you really negate freezing point depression or is it only a diminution?

Table 1: The word "initial" can be confusing here as it describes the cell concentrations after culturing. Please optimize caption and footnote.

Table 2: I would recommend to explain "n/a" in a footnote rather than in the caption.

Fig. 1: Why do you have sometimes the same frozen fraction for different temperatures (also in the other figures)? I would recommend to show only changes in the frozen fraction. Please explain.

Fig. 2: You describe the plain lines in panel (b) as spectra for media controls, which underwent the heat and peroxide treatments. I can only see one plain line in panel (b) additional to the sample spectrum, but two more dashed lines, which have not been described in the caption yet. Please clarify the different lines. I would also recommend to use symbols instead of only colors in the legend. In the legend, you write "peroxide" but in the caption you name it "H2O2". Please be consistent. What kind of error bars did you use here? How do you explain that the heat-treated medium control shows a higher IN activity compared to the unamended control (panel b)?

Fig. 3: Please add a legend to the figure. Please also consider additional comments

of Fig. 2. In this figure, it is the other way around, the peroxide-treated medium control shows a higher IN activity compared to the unamended control. Please explain.

It would be beneficial for the community to calculate an INP concentration using Vali's equation and compare the results to other biological INP.

―――――――――――――――――――

---

## Referee Comment (RC2) · Anonymous Referee #2 · 4 Jan 2021

There have been no data available on the ice nucleating activities of archaea, and this study examined the capacity of four haloarchaeal species with different cell wall types to serve as INPs. In general, the cells that remained intact after dilution in distilled water incited freezing at the warmest subzero temperatures observed, and additional experiments provided evidence that the activity is mediated by a proteinaceous or organic compound associated with the cells. I suggest a minor revision for the title by replacing "haloarchaea" for archaea since that was the only type of archaeal species they tested and more accurately describes the study. Below are more detailed and specific comments to consider when revising this manuscript.

Abstract, Lines 19-22: As written, this sentence implies that thermophiles are prevalent "in other cold niches", which is not where one might expect thermophiles to be

prevalent.

Abstract, Lines 22-24: Since the ability of archaea to "become airborne" or "impact cloud formation" was not examined in this study, please consider revising this closing statement in the abstract to more directly reflect the results obtained and their implications.

Line 74: Suggest revising this statement to "up to 40% of the microbial taxa in an ecosystem".

Lines 118-119; 132-133: That fresh cultures were sent overnight is described, but please indicate how much time passed between receiving the cultures at CSU and performing the ice nucleation assays. This is very important information needed to evaluate the results because it is well established that the phase of growth and culture age affect ice nucleation activity in bacteria (e.g., Nemecek-Marshall et al. 1993, J Bacteriol 175:4062–4070; Fall and Fall 1998, Curr Microbiol 36:370–376; Yankofsky et al. 1983, Current Microbiology 9:263–267).

Lines 122-123: This sentence describes a result and is out of place in the methods.

Lines 124-126; 147-148: These sentences in the methods would be more appropriate for the discussion section.

Line 128: Please clarify what is meant by "active". Do you mean ice nucleation active? Metabolically active?

Lines 145-147: Please provide more detail on how the UV exposure was done to irradiate the liquid samples. If this was done by exposing a sample held within a test tube and since UV is opaque to most plastics as well as being attenuated by water and particulates (cells), it is important to explain the composition of materials involved and procedure in more detail (e.g., dose rate of UV source, distance of samples from the source, and if the dense suspension was mixed during exposure). Please also indicate the final concentration of peroxide used in the experiments in v/v. For example, if 0.75

mL of 30% H2O2 + 1.5 mL aliquot of suspension = 2.25 mL, so was it 7.5% H2O2 v/v? Finally, are all the haloarchaea used in this study catalase negative?

Line 163: Please clarify what is meant by "markers" and note that polysaccharides are ubiquitous components of archaeal and bacterial cell envelopes.

Lines 168-169; 171-172: Please consider revising line 168 to state that "lysed cells" of these species do not have ice nucleation activity. The authors should also consider mentioning that it is well known that lysing of ice nucleation active bacterial cells decreases the efficiency at which they are INPs (e.g., Lindow et al. 1989, Mol. Plant-Microbe Interact. 2, 262). Are there any data available from experiments with lysed cells of Halococcus morrhuae and Haloferax sulfurifontis? It would not be surprising if the lysed cells

Lines 176-179: Figure 2 indicates that INPs active at the warmest temperatures were heat labile, so I'm confused by what is meant by a "more substantial amount" of something else. Since the fraction of samples that froze at each temperature is known, this can be used to calculate the number of INPs at each temperature according to the method of Vali (1971, J Atmos Sci 28:402–409). These data provide context for inferring the fraction of the cell populations that were ice nucleation active at a given temperature/experimental condition.

Lines 182-184: When catalase was added to samples of the less dilute cell suspensions, were oxygen bubbles observed/produced? I follow this argument, but it has me wondering about the "residual organic material" statement. Are the authors suggesting that treatment of the cell suspensions with peroxide oxidizes all macromolecules and organic constituents of the cells completely to $CO_2$?

Lines 187-188: I think this section is talking about Figure 3, but on closer inspection, I don't see Figure 3 referred to in the main text.

Lines 188-190: Please explain how these different behaviors should be interpreted with

respect to the properties that can be inferred from the archaeal INPs.

Lines 198-199: I would not describe the salt concentrations used in these experiments as "low", at least not in comparison to rain, snow, or freshwaters. The average concentration of salt used in the assays was ∼1% and is roughly a quarter seawater. Please can the authors describe conditions that would allow cloud droplets to achieve such high ionic strength.

Lines 205-206: Please expand on this point as I am not aware of any work that has shown similarities in motifs between gammproteobacterial IN proteins and S-layer proteins.

Figure 3: not mentioned in main text.

---

## Referee Comment (RC3) · Anonymous Referee #3 · 15 Jan 2021

Ice nucleation activity of biotic material is an interesting topic and investigation ice nucleating properties pf Archaea is an interesting approach. However, the problem with this study is that I do not feel that there was well thought out experimental design. 1. The authors did not treat all four organisms the same, comparing different dilutions, intact cells from one species vs lysed cells from another. 2. The authors argue correctly that, once airborne, halophiles would be exposed to a dilute environment. However, the salinity of the selected dilution is not representative of cloud droplets. Moreover, why intact cells that did show ice nucleation activity where not further diluted to cause cell lysis, and conversely, cells that were lysed were not tested at lower dilutions to keep cells intact is unclear. 3. The authors reported that diluting the media reduced survivability of halophiles and, in order to account for that, they determined the number

of intact cells by microscopy. Yet, cell numbers reported in Table 1 are simply derived by multiplying cell numbers by the dilution factor (Table 2), and thus, the study does not account for any losses caused by the dilutions. Undiluted cell suspensions should have been fixed to avoid cell lysis and counted. How can the authors account for a combined effect of intact cells and lysed cell material in these assays? 4. The purpose of 30% H2O2 treatment is to determine the contribution of abiotic factors. However, the authors question the efficiency of the digestion protocol. If these 'digests' are a mixture of biotic and abiotic compounds, then there is no need for including the data. 5. There is no statistical evidence presented that any of the organisms/treatments/controls were different albeit 24 replicates.

As intact cells for H. walsbyi and N. pharaonic were not investigated and lysed cells did not show ice nucleation activity, they do not contribute to the study and should be removed.

Methodology: L116: It is confusing and unclear why it is relevant the cells were first grown at the College of Charleston and then shipped to Colorado State University. L116: Unclear why cells were first grown to mid-log phase but subsequently to somewhere during log-phase. L117: Please provide more detail no monitoring cell density. What microscope, cell counts, add reference on microscopy-cell abundance procedure. Also, the objective was most likely a 100 x with an extra 10x magnification within the eyepiece or camera. Unclear why cell density would be of relevance prior to shipping? L117: reference to Table 1: cells were shipped to a different university and additional experiment where performed at this university I assume. Cell numbers always remained constant during shipping, storage, and time passed until experimental setup? Table 1: Is salinity presented as gram NaCl? I am asking because e.g. DSMZ media 97 contains 250 g NaCl. Should this be 250 ppt and 25%. L127: If the current study truly determined the lysis thresholds, why did the authors not in include dilutions for H. walsbyi and N. pharaonic. Cells were diluted 1:15 one time not serially. Dilution resulted in the lysis of two out of four selected organisms. Reason why not includ-

Interactive
comment

ing lower dilutions that left cells intact as well as including dilutions for the other two organisms that would have resulted in cell lysis is unclear; particularly since the authors determined lysis thresholds. L128: Why were cells grown again if they already reached the desired cell density prior to shipping (L117)? Ll29: What was the reason for selecting two more dilutions for H. morrhuae but not the other three organisms? It is well known that cell density has an effect on ice nucleation. L130: add detail on microscopy. L138: what was the coolant? L142-148: (i) Heat treatment and 30% H2O2 amendments were intended to determine the effect lysed cell material and inorganic molecules on ice nucleation. If the authors think that the digestion was ineffective, they should have altered the protocol rather than hinting at the need for it. (ii) a general problem throughout the experimental design is the comparison of different cell 'material' i.e., intact, lysed, intact/digested, lysed/digested. It appears that the initial intact vs. lysed was an artifact and the authors went with it, but essentially, the study compares apples and oranges.

Results and discussion As the results and discussion section will largely change after removing a large portion of the dataset here are some general comments. When did controls freeze? H. morrhuae is atypical compared to all Archaea or the ones investigated? L167-172: Irrelevant as lyses cells are compared to intact cells. This is a study on INPs. No sure why it is 'interesting or relevant' to discuss cell lyses. As mentioned before low dilutions should have been used to not lyse cells. L176-175: H2O2 treatment should provide information in the abiotic fraction not organic. L174-: Are any of these reported changes in freezing temperatures statistically different from intact cells? L183-186: Discussion on ineffective H2O2 treatment. Effectively separating or removing specific fractions when investigating ice nucleation properties is essential. This section does not strengthen the manuscript. L192-193: Why is N. pharaonis now only partially lysed contradicting previous statements?

Conclusion: L209-213: seems more suited for an introduction.

Figure 1: Please remove the shading. The reader gets the impression that the area

under the curve is of importance. Why on a log scale?

Figure 2: Why are there no controls for the dilutions 1:6 and 1:30? As shown in Table 2 the salinity ranges from 2.7%-0.5% in these samples.

Other edits: Check for missing punctuations L10: delete 'from microorganisms' L15-16: change 'of a subset of archaeal cells from Haloarchaea' to 'selected genera of the class Haloarchaea' L16-18. Reason for comparing intact cells two lysed cells from different genera is unclear L17: without comparing to the freezing temperature of an abiotic control, I would not consider -18C warm. Please rephrase. L18: What are warm temperature INPs? Please rephrase. L23: 'necessary to improve'. These are extremely strong words. How about 'intriguing' L40: delete ' approximately' L54: replace 'however' with 'further' L57-61: please split this running sentence. L64: Do all minerals except for feldspars function as INPs? I suggest deleting 'aside some feldspars' L71: delete 'relatively' L82-83: Are there some bacterial cell that produce no peptidoglycan as the authors say 'nearly ubiquitous'? L93: 'possess' L96: replace 'it is' with 'its' L97: For the other three genera cell characteristics were briefly described as a justification to include them in this study. Why not for Natronomonas? L100: change to 'they are relatively easy to culture compared to other archaeal lineages' L105: What do the authors mean by 'halophiles and hypersaline' L106-109: I recommend removing this paragraph. This paragraph is trying to oversell the importance of this research. L116: delete '(i.e., midway through the period of exponential cell growth)' L118: delete '(i.e., the period characterized by cell doubling)' L141: delete 'for all species during each experiment.'

---

## Author Comment (AC2) · 28 Feb 2021

*We would like to thank the reviewers for their valuable comments. We have revised the manuscript accordingly and think it has strengthened as a result. Please find our responses to reviewer comments and changes to the manuscript below in blue text. A track changes version is also included.*

**Reviewer 1**

General comments:

Creamean at al. studied ice nucleation activity within the domain Archaea and are the first to report ice nucleation activity of this domain. Up to date, Archaea have not been evaluated as INP. In two out of four investigated species, the ability to induce freezing above -18 ◦ C was found. The authors performed additional experiments (heat treatment, peroxide digestions) to further study the composition of INPs from Archaea. They suggest that the IN activity of intact cells were driven by organic and heat-labile materials. This work provides valuable insights into ice nucleation activity within the domain Archaea and contributes to the understanding of biological ice nucleation as a whole. The results are certainly publishable and are aligned with the scope of the journal. Overall, the manuscript is well-written and -structured, and I recommend to publish it in Biogeosciences after the following comments have been addressed.

Specific comments:

Line 111: Where did you obtain the cell cultures from? Please provide more information.

*All cultures were purchased from the Leibniz Institute DSMZ-German Collection of Microorganisms and Cell Cultures. This information has been included in section 2.1.*

Line 117: How did you determine the salinities?

*Salinities were confirmed using a handheld refractometer. This information has been added to section 2.1.*

Line 122, Table 1, Table 2: Usually ppt is understood as "parts per trillion". In specific disciplines such as oceanography, however, the abbreviation ppt is commonly used for "parts per thousand". I would recommend additional annotations to minimize misunderstandings.

*Thank you for bringing this to our attention. We have clarified in the footnotes for both tables that ppt stands for "parts per thousand".*

Line 137: What is the temperature uncertainty of the measurement?

*The temperature uncertainty of the IS is less than ±0.2 °C, which is a combination of the uncertainty in the thermocouples and the temperature variation across the blocks due to gradients in cooling. We have added this information to section 2.3.*

Line 138: If the freezing of wells was recorded in 0.5 ◦ C steps, how did you receive initial freezing temperatures such as -17.6 ◦ C or -19.2 ◦ C (lines 152-153)? Why do you have often more than two data points within one degree in all figures?

*Thank you for pointing this out. We ended up recording and plotting all data points (i.e., at approximately 0.33 °C intervals based on the cooling rate). We have clarified this in section 2.3.*

Line 140: How many independent experiments have been performed? Have the results been checked for reproducibility? Are the data presented in the figures arithmetic mean values? Otherwise, how did you calculate the standard error?

*Independent experiments were not performed in this initial survey. As the reviewer implies, growth at different temperatures and, especially, on/in different media can significantly affect ice nucleation activity of the well-known species of Gram–ve ice nucleation active (INA) bacteria (e.g., Nemecek-Marshall et al. 1993; Ruggles et al. 1993). With the single known species of bacterium from the Gram +ves, a Lysinibacillus sp., growth on two different media shifted the spectrum by 1 – 3 °C (Failor et al. 2017). Our experience with both INA bacteria and fungi has shown us that growth conditions affect activity within a limited temperature range; more important is the intrinsic ability of an organism to fabricate INA molecules, which tend to have limits over which they activate. A constraint with these haloarchaeal species is that we were limited in choices of media due to their specific growth requirements. In this study we did not find evidence that the isolates were capable of exceptional INA ability (i.e., ≥ –10 °C). If they had, we would have conducted further experiments to explore the effect of growth conditions and salinity upon it.*

*Failor, K.C. et al. 2017. Ice nucleation active bacteria in precipitation are genetically diverse and nucleate ice by employing different mechanisms. The ISME journal, 11, 2740-2753.*

*Nemecek-Marshall, M. et al. 1993. High level expression of ice nuclei in a Pseudomonas syringae strain is induced by nutrient limitation and low temperature. J. Bacteriol. 175, 4062–4070.*

*Ruggles, J. A., et al. 1993. Kinetics of appearance and disappearance of classes of bacterial ice nuclei support an aggregation model for ice nucleus assembly. J. Bacteriol. 175, 7216–7221.*

*In order to be consistent with our previously published IS data, we now use 95% confidence intervals calculated based on the methodology of Agresti and Coull (1998). This is now clarified in the methods and captions.*

Line 142: For simplification, I would recommend to use the terms "heat" and "peroxide" (treatment) throughout the whole manuscript as they are also used in the figure legends.

*Done.*

Line 172: Is there any proof that the intact cells of H. walsbyi and N. pharaonis would show ice nucleation ability, which can be suppressed by lysis? You could additionally test if the ice nucleation ability of H. morrhuae and H. sulfurifontis can be suppressed upon lysis.

*Unfortunately, it is not possible to test the ice nucleation activity of H. walsbyi and N. pharaonis intact cells because they are particularly sensitive to lysis under hyposaline conditions (e.g., Boring et al., 1963). They readily lyse in water roughly 5% NaCl and below. The dilutions required to maintain their intact nature would still result in saline conditions that would significantly depress the freezing point of water. H. walsbyi and N. pharaonis require such high salinities in order not to lyse that the depression would be approaching the lower limit of the IS, so we would not obtain much of a spectrum. This is based on control tests using 10% saline solution controls with artificial seawater (Instant Ocean® Sea Salt) and solutions with archaea whereby freezing would not be observed until < –23 °C and then only reach ~ 0.2 fraction frozen at the IS lower limit. So, we would not have much of a spectrum left if using higher salinities to enable these archaea to remain intact.*

*We cannot test with H. morrhuae because it does not readily lyse in fresh water and is difficult to lyse without significant treatments that would alter its cellular properties (Leuko et al., 2004). A sufficient dilution would result in most H. sulfurifontis cells lysing but to be consistent with our methodologies with the other species would have resulted in a sample with extremely low cell density.*

*We have added Leuko et al. (2004) to the introduction where we stated that Halococcus do not lyse in fresh water. We added a statement on the particularly sensitive behavior of H. walsbyi and N. pharaonic to saline conditions to the introduction. We also added the rationale above for the chosen dilutions to the methods.*

*Boring, J., Kushner, D. J., and Gibbons, N. E.: Specificity of the salt requirement of halobacterium cutirubrum, Canadian Journal of Microbiology, 9, 143-154, 10.1139/m63-020, 1963.*

*Leuko, S., Legat, A., Fendrihan, S., and Stan-Lotter, H.: Evaluation of the LIVE/DEAD BacLight Kit for Detection of Extremophilic Archaea and Visualization of Microorganisms in Environmental Hypersaline Samples, Appl Environ Microbiol, 70, 6884, 10.1128/AEM.70.11.6884-6886.2004, 2004.*

Lines 177, 180-182, 187-188: Which temperature do you refer to? Initial freezing temperature? T50?

*These refer to the decreases in temperature when comparing the treatments to the standard INP spectra for initial freezing temperatures and when samples reached 100% frozen (i.e., final freezing temperatures). Although, we realize this is confusing. We have changed this so that these decreases (see new values in section 3.2) represent the average drop in temperature from the unamended INPs to the treatments (i.e., by calculating the average freezing temperature over the course of each spectrum and subtracting from the unamended). We have clarified this at the beginning of section 3.2 as well.*

Lines 182-186: Have you tried to use an excess of peroxide to remove all organic material, which could serve as INP, so that there are no differences between sample concentrations anymore?

*Based on results from many previous studies, we have found that digestion in 10% $H_2O_2$ provides an excess of oxidant for full decomposition of susceptible organic material. But the reviewer makes a very good point. Very occasionally we see residual activity of warm INPs in atmospheric aerosol samples, which suggests the presence of a population of INPs that are resistant to oxidation (note that we know from the amount of catalase required to neutralize excess peroxide that it had not been used up during the process). With a couple of these samples, we have tried doubling the time of the digestion, as shown at right. We found that this reduced the residual INP population,*

[Figure]

*but did not remove it altogether, suggesting that this residual "hump" was either protected from oxidation or resistant to it (e.g., soot is not oxidized by $H_2O_2$).*

*Perhaps the archaea organic INPs are unique and some may be embedded within entities and are somewhat recalcitrant, making it difficult for the peroxide to completely degrade. We hope that any*

*future investigations of archaeal INPs would include a more rigorous peroxide treatment regimen, akin to that of the recent Bogler et al. (2020) study on lignin biopolymer. We added a statement on our recommendation after this sentence in the revision and included a statement with references on the basis for choosing the amount of peroxide used.*

Lines 187-188: Please refer to Figure 3 here.

*Done.*

Lines 190-191: Please clarify: "Interestingly, H. sulfurifontis INP spectra were more responsive to the heat treatment as opposed to peroxide, . . .". Assuming that the same symbols and gray colors were chosen for figures 2 and 3 (legend is missing), the peroxide treatment shows a stronger reduction of INP spectra than the heat treatment.

*We did mix up the symbols and colors and have fixed this. For clarity, we added a legend to Figure 3 as well. This statement is accurate; however, we did revise this paragraph to make this clear. H. morrhuae which was more responsive (i.e., had a larger drop in average freezing temperature) to peroxide than heat while H. sulfurifontis was the opposite in that there was a greater degradation with heat alone than was further enhanced with the addition of peroxide.*

Line 197: Can you really negate freezing point depression or is it only a diminution?

*Probably cannot completely negate but can reduce to negligible effects. We changed "negate" to "reduce".*

Table 1: The word "initial" can be confusing here as it describes the cell concentrations after culturing. Please optimize caption and footnote.

*We removed "initial".*

Table 2: I would recommend to explain "n/a" in a footnote rather than in the caption.

*Done.*

Fig. 1: Why do you have sometimes the same frozen fraction for different temperatures (also in the other figures)? I would recommend to show only changes in the frozen fraction. Please explain.

*This is because no new drops froze and thus the fraction frozen remained the same between temperatures. However, we understand this may be confusing and have removed redundant data with the same fraction frozen for different temperatures.*

Fig. 2: You describe the plain lines in panel (b) as spectra for media controls, which underwent the heat and peroxide treatments. I can only see one plain line in panel (b) additional to the sample spectrum, but two more dashed lines, which have not been described in the caption yet. Please clarify the different lines. I would also recommend to use symbols instead of only colors in the legend. In the legend, you write "peroxide" but in the caption you name it "H2O2". Please be consistent. What kind of error bars did you use here? How do you explain that the heat-treated medium control shows a higher IN activity compared to the unamended control (panel b)?

*We meant lines without markers, which we now have updated the caption to reflect this. For clarity, we now provide more comprehensive legends in all figures. We have changed $H_2O_2$ to "peroxide" and added a statement about the error bars (95% confidence intervals).*

Fig. 3: Please add a legend to the figure. Please also consider additional comments of Fig. 2. In this figure, it is the other way around, the peroxide-treated medium control shows a higher IN activity compared to the unamended control. Please explain.

*Done.*

It would be beneficial for the community to calculate an INP concentration using Vali's equation and compare the results to other biological INP.

*We originally considered this; however, this is trickier for haloarchaea versus other microorganisms. Given some samples were subject to lysing (H. walsbyi and N. pharaonic, which would under any hyposaline conditions) and some were not (H. morrhuae and H. sulfurifontis), comparing INPs mL$^{-1}$ of sample, with the inherent assumption that this equate to INPs per cell, for intact versus fragmented cells would be comparing apples to oranges. With the uncertainties introduced by cell lysing, we ultimately kept it as fraction frozen for a more straightforward comparison. Fraction frozen is still shown on occasion (e.g., Adams et al., 2021). We could have calculated the number of active sites like Adams et al. but since we have the issue of intact versus lysed cells in our dilutions, this would still not be realistic when intercomparing the different haloarchaea.*

*Adams, M. P., Atanasova, N. S., Sofieva, S., Ravantti, J., Heikkinen, A., Brasseur, Z., Duplissy, J., Bamford, D. H., and Murray, B. J.: Ice nucleation by viruses and their potential for cloud glaciation, Biogeosciences Discuss. [preprint], https://doi.org/10.5194/bg-2020-474, in review, 2021.*

*However, we did like the idea of comparing to other biologically-derivedun INPs, so we have added a figure (Figure 4) that summarizes reported freezing temperature ranges for known biologically-derived INPs and discuss briefly in section 4.*

**Reviewer 2**

There have been no data available on the ice nucleating activities of archaea, and this study examined the capacity of four haloarchaeal species with different cell wall types to serve as INPs. In general, the cells that remained intact after dilution in distilled water incited freezing at the warmest subzero temperatures observed, and additional experiments provided evidence that the activity is mediated by a proteinaceous or organic compound associated with the cells. I suggest a minor revision for the title by replacing "haloarchaea" for archaea since that was the only type of archaeal species they tested and more accurately describes the study. Below are more detailed and specific comments to consider when revising this manuscript. Abstract, Lines 19-22: As written, this sentence implies that thermophiles are prevalent "in other cold niches", which is not where one might expect thermophiles to be prevalent.

Abstract, Lines 22-24: Since the ability of archaea to "become airborne" or "impact cloud formation" was not examined in this study, please consider revising this closing statement in the abstract to more directly reflect the results obtained and their implications.

*Done. We changed this sentence to, "Thus, it is important to assess their ability to serve as INPs as it may lead to an improved understanding of biological impacts on clouds."*

Line 74: Suggest revising this statement to "up to 40% of the microbial taxa in an ecosystem".

*Done.*

Lines 118-119; 132-133: That fresh cultures were sent overnight is described, but please indicate how much time passed between receiving the cultures at CSU and performing the ice nucleation assays. This is very important information needed to evaluate the results because it is well established that the phase of growth and culture age affect ice nucleation activity in bacteria (e.g., Nemecek-Marshall et al. 1993, J Bacteriol 175:4062–4070; Fall and Fall 1998, Curr Microbiol 36:370–376; Yankofsky et al. 1983, Current Microbiology 9:263–267).

*We ended up removing the information on institutions where the cultures were grown and shipped to as that seemed irrelevant but did change the last sentence of section 2.1 to, "Cultures were measured for their ice nucleation abilities within 1 – 2 days upon achieving log-phase growth."*

Lines 122-123: This sentence describes a result and is out of place in the methods.

*We think this is important to include here because in general, salinity reductions can cause haloarchaea to lyse. We revised the sentence to make this clear in that it is an effect that is more broadly applicable to haloarchaea: "Reductions in salinity can inherently cause the cells of certain haloarchaeal species to lyse (Boring et al., 1963; Legat et al., 2010; Leuko et al., 2004)."*

Lines 124-126; 147-148: These sentences in the methods would be more appropriate for the discussion section.

*We moved the sentence from lines 124-126 to the end of section 3.1: "Testing ice nucleation responses from cell lysis is relevant given: (1) cell fragments from fungi and bacteria have been previously observed to serve as INPs (Anderson and Ashworth, 1986; Du et al., 2017; O'Sullivan et al., 2015; Šantl-Temkiv et al., 2015) and (2) a less saline environment is more atmospherically relevant for how archaea might behave once incorporated in a relatively dilute cloud drop prior to immersion freezing. However, the results presented here indicate that of the species studied, lysed haloarchaeal cells (i.e., cell fragments) do not enhance ice nucleation abilities, and possibly even suppress it."*

*We did not move the sentence on lines 147-148 as that is based on previous work but did change the tense to reflect that and added a couple of references.*

Line 128: Please clarify what is meant by "active". Do you mean ice nucleation active? Metabolically active?

*Changed to "metabolically active".*

Lines 145-147: Please provide more detail on how the UV exposure was done to irradiate the liquid samples. If this was done by exposing a sample held within a test tube and since UV is opaque to most plastics as well as being attenuated by water and particulates (cells), it is important to explain the composition of materials involved and procedure in more detail (e.g., dose rate of UV source, distance of samples from the source, and if the dense suspension was mixed during exposure). Please also indicate the final concentration of peroxide used in the experiments in v/v. For example, if 0.75 mL of 30% $H_2O_2$ + 1.5 mL aliquot of suspension = 2.25 mL, so was it 7.5% $H_2O_2$ v/v? Finally, are all the haloarchaea used in this study catalase negative?

*Reviewer 2 raises a very valid point. In designing this method, we knew that most suspensions of environmental media already contain species such as manganese dioxide or $Fe^{3+}/Fe^+$ to drive the production of hydroxyl radicals. However, we reasoned that ensuring their production was sensible. We knew that UVC was blocked by glass and plastics, so we chose UVA/UVB-generating bulbs designed to*

*penetrate terrariums with plastic or glass covers. Efficiency of hydroxyl radical production by UVA/UVB is much lower than with UVC, so we use two 26 W bulbs placed adjacent to the floating samples, as sown. The suspension was not dense, since as noted we used the diluted samples, and were mixed by being agitated by boiling and the large stirrer bar, as shown. We have added the following details and modified the methods to provide more details, thus: "while illuminated with UVA/UVB fluorescent bulbs (Exo Terra Reptile UVB, 2 × 26 W providing ~2,000 µW cm$^{-2}$ UVA and ~300 µW cm$^{-2}$ UVB at the distance used)."*

[Figure]

[Figure]

*With reference to the potential production of catalase by the isolates, this is not an issue of concern since immediately after addition of the aliquot of 30% H$_2$O$_2$ we immerse the samples in boiling water, which would rapidly denature any catalases, if present.*

Line 163: Please clarify what is meant by "markers" and note that polysaccharides are ubiquitous components of archaeal and bacterial cell envelopes.

*Meant that they have been used as tracers for INPs as demonstrated in Zeppenfeld et al. (2019). We changed "markers" to "tracers".*

Lines 168-169; 171-172: Please consider revising line 168 to state that "lysed cells" of these species do not have ice nucleation activity. The authors should also consider mentioning that it is well known that lysing of ice nucleation active bacterial cells decreases the efficiency at which they are INPs (e.g., Lindow et al. 1989, Mol. PlantMicrobe Interact. 2, 262). Are there any data available from experiments with lysed cells of Halococcus morrhuae and Haloferax sulfurifontis? It would not be surprising if the lysed cells.

*Thank you for bringing this to our attention. We changed the first sentence to, "Lysed cells of both H. walsbyi and N. pharaonis did not exhibit ice nucleation activity...". It is true that lysing the well-known INA bacteria greatly lowers their ice nucleation activity. However, this is because their IN proteins are embedded within their outer membrane, where they also agglomerate together, increasing the INA. These archaea are from an entirely different kingdom and do not possess outer membranes. Nor do we know if the INA metabolites are proteins. Hence, we think it would be premature to compare their mechanism of activity with the uniquely-active Gram –ve group, that includes P. syringae.*

*There are no data of lysed cells of H. morrhuae because these do not lyse in fresh water. A sufficient dilution would result in most H. sulfurifontis cells lysing but to be consistent with our methodologies with the other species would have resulted in a sample with extremely low cell density.*

*We added the following sentence to the end of section 3.1: "This is analogous to previous work on bacteria, whereby it is well known that lysing of INA bacterial cells decreases the efficiency at which they are INPs (e.g., Lindow et al., 1989)."*

Lines 176-179: Figure 2 indicates that INPs active at the warmest temperatures were heat labile, so I'm confused by what is meant by a "more substantial amount" of something else. Since the fraction of samples that froze at each temperature is known, this can be used to calculate the number of INPs at each temperature according to the method of Vali (1971, J Atmos Sci 28:402–409). These data provide context for inferring the fraction of the cell populations that were ice nucleation active at a given temperature/experimental condition.

*Even changes in fraction frozen can indicate a more substantial amount (here, for comparing heat-labile versus organic materials), but it is relative since we are talking about fraction frozen. Thus, we changed to, "These results indicate the samples contained some heat-labile, likely proteinaceous, INPs, but contained a relatively larger contribution from other biogenic organic INPs..."*

*We originally considered calculating INP concentrations using Vali (1971); however, this is trickier for haloarchaea versus other microorganisms. Because we do not have cell concentrations for all dilutions due to lysing, we could not calculate it as something like INP concentration per mL of cell suspension. Given some samples were subject to lysing (H. walsbyi and N. pharaonic, which would under any hyposaline conditions) and some were not (H. morrhuae and H. sulfurifontis), comparing INP concentrations for intact versus fragmented cells would be comparing apples to oranges. This was why we ultimately kept it as fraction frozen: to be able to compare all the species together. Fraction frozen is still shown on occasion (e.g., Adams et al., 2021). We could have calculated the number of active sites like Adams et al. but since we have the issue of intact versus lysed cells in our dilutions, this would still not be realistic when intercomparing the different haloarchaea.*

*Adams, M. P., Atanasova, N. S., Sofieva, S., Ravantti, J., Heikkinen, A., Brasseur, Z., Duplissy, J., Bamford, D. H., and Murray, B. J.: Ice nucleation by viruses and their potential for cloud glaciation, Biogeosciences Discuss. [preprint], https://doi.org/10.5194/bg-2020-474, in review, 2021.*

Lines 182-184: When catalase was added to samples of the less dilute cell suspensions, were oxygen bubbles observed/produced? I follow this argument, but it has me wondering about the "residual organic material" statement. Are the authors suggesting that treatment of the cell suspensions with peroxide oxidizes all macromolecules and organic constituents of the cells completely to CO2?

*Yes, they were, liberally. We find that using ≥10% $H_2O_2$ provides a good excess of oxidant, and that even with "dirty" samples, such as permafrost suspensions, the great bulk of the peroxide remains after the*

*digestion, requiring the full aliquot (90 μL) of catalase to neutralize it. As mentioned above, in response to Reviewer 1, we do sometimes find aerosol samples that appear to have organic INPs that are resistant to oxidation. Soot and plastics, for example, are not affected by $H_2O_2$, and organic INPs may be protected by adsorption onto minerals or by being coated by clay particles.*

Lines 187-188: I think this section is talking about Figure 3, but on closer inspection, I don't see Figure 3 referred to in the main text.

*We have added a reference to Figure 3 in the last paragraph of section 3.2.*

Lines 188-190: Please explain how these different behaviors should be interpreted with respect to the properties that can be inferred from the archaeal INPs.

*We have added the following: "Collectively, these results indicate that H. morrhuae contained more organic relative to heat-labile INPs, while H. sulfurifontis contained more heat-labile as opposed to organic INPs. These haloarchaea have very different cellular envelop compositions: H. sulfurifontis contains a proteinaceous S-layer while H. morrhuae is devoid of such an S-layer but instead possessed a cell envelope that is composed of highly sulphated heteropolysaccharides. Thus, it would make sense that H. sulfurifontis is more sensitive to heat than peroxide given its proteinaceous cell envelope (assuming those proteins are ice nucleation active) and H. morrhuae is more sensitive to peroxide than heat given its polysaccharide-rich cell envelope."*

*Thank you very much for leading us to examine this more closely, as the cellular structure of each can help explain their responses to the treatments, which is something we did not realize before!*

Lines 198-199: I would not describe the salt concentrations used in these experiments as "low", at least not in comparison to rain, snow, or fresh waters. The average concentration of salt used in the assays was ~1% and is roughly a quarter seawater. Please can the authors describe conditions that would allow cloud droplets to achieve such high ionic strength.

*Good point. We have omitted the latter half of this sentence and just specified that we used hyposaline conditions to reduce freezing point depression, which was the primary objective of the dilutions.*

Lines 205-206: Please expand on this point as I am not aware of any work that has shown similarities in motifs between gammproteobacterial IN proteins and S-layer proteins.

*This point is based on the fact that we found proteins with the domain DUF3494 which has been commonly shown to be involved in ice-binding to be present in at least two haloarchaeal species. This is now explained in section 4.*

*Ice-binding ability is a characteristic of both INA and antifreeze proteins. Indeed, as Eickhoff et al. (2019) explain, "It has been proposed that both types of proteins interact similarly with ice and that, in principle, they may be able to exhibit both functions. We show that in addition to ice growth inhibition, both can also trigger ice nucleation..., providing unambiguous experimental proof for their contrasting behavior. Our analysis suggests that the predominant difference between AFPs and INPs is their molecular size, which is a very good predictor of their ice nucleation temperature." See also Kobashigawa et al. (2005) and Qiu et al. (2019). We have added the following to the conclusions: "Ice-binding ability is a characteristic of both ice nucleating and antifreeze proteins and is influenced primarily by their size (Eickhoff et al., 2019; Qiu et al., 2019).*

Eickhoff, L., Dreischmeier, K., Zipori, A., Sirotinskaya, V., Adar, C., Reicher, N., Braslavsky, I., Rudich, Y. and Koop, T. 2019. Contrasting behavior of antifreeze proteins: Ice growth inhibitors and ice nucleation promoters. The Journal of Physical Chemistry Letters, 10, 966-972.

Kobashigawa, Y., Nishimiya, Y., Miura, K., Ohgiya, S., Miura, A. and Tsuda, S. 2005. A part of ice nucleation protein exhibits the ice-binding ability. FEBS Lett., 579, 1493-1497.

Qiu, Y., Hudait, A. and Molinero, V., 2019. How size and aggregation of ice-binding proteins control their ice nucleation efficiency. Journal of the American Chemical Society, 141, 7439-7452.

Vance, T. D. R., Bayer-Giraldi, M., Davies, P. L., and Mangiagalli, M.: Ice-binding proteins and the 'domain of unknown function' 3494 family, Febs j, 286, 855-873, 10.1111/febs.14764, 2019.

Figure 3: not mentioned in main text.

*We now explicitly mention it when we discuss the results from this figure in the last paragraph in section 3.2.*

**Reviewer 3**

Ice nucleation activity of biotic material is an interesting topic and investigation ice nucleating properties pf Archaea is an interesting approach. However, the problem with this study is that I do not feel that there was well thought out experimental design.

1. The authors did not treat all four organisms the same, comparing different dilutions, intact cells from one species vs lysed cells from another.

*We actually did treat them all the same. We conducted a 1:15 dilution for all the haloarchaeal species (see Table 2 and Figure 1), the results from which are used when intercomparing all four together. We also subjected all four to the treatments, but as we stated on lines 192-193, H. walsbyi and N. pharaonis showed no response to the treatments because there were little to no INPs in the original dilutions and thus we did not show them in a figure or discuss them further. We added a note of this in the beginning of section 3.2. Given only H. morrhuae and H. sulfurifontis exhibited ice nucleation activity above the media controls, we did show and discuss results from their treatments. Since H. morrhuae was the most active (Figure 1), we further tested it under various dilutions to assess if the activity changes. It is reasonable to conduct different tests on different species given their activities and only intercompare species for the same testing. We did not intercompare species under different testing conditions.*

*As we indicated in the introduction and, as is generally known, not all microorganisms harbor the same properties. For example, not all bacteria are efficient INPs like P. syringe. These archaea have very different biological makeup, and surface properties and shapes, thus, would behave differently under variable conditions. Naturally, as with other microorganisms like bacteria, certain ones lyse and certain ones do not under hyposaline conditions (e.g., Lindow et al., 1989). The same applies to these haloarchaeal species. However, testing them in their hypersaline suspensions to ensure they remained intact would not make sense as that would significantly decrease the freezing point of water and would not be environmentally representative.*

*Unfortunately, it is not possible to test the ice nucleation activity of H. walsbyi and N. pharaonis intact cells because they are particularly sensitive to lysis under hyposaline conditions (e.g., Boring et al., 1963). They readily lyse in water roughly 5% NaCl and below. The dilutions required to maintain their intact nature would still result in saline conditions that would significantly depress the freezing point of*

*water. H. walsbyi and N. pharaonis require such high salinities in order not to lyse that the depression would be approaching the lower limit of the IS, so we would not obtain much of a spectrum. This is based on control tests using 10% saline solution controls with artificial seawater (Instant Ocean® Sea Salt) and solutions with archaea whereby freezing would not be observed until < –23 °C and then only reach ~ 0.2 fraction frozen at the IS lower limit. So, we would not have much of a spectrum left if using higher salinities to enable these archaea to remain intact.*

*We cannot test with H. morrhuae because it does not readily lyse in fresh water and is difficult to lyse without significant treatments that would alter its cellular properties (Leuko et al., 2004). A sufficient dilution would result in most H. sulfurifontis cells lysing but to be consistent with our methodologies with the other species would have resulted in a sample with extremely low cell density.*

*We have added Leuko et al. (2004) to the introduction where we stated that Halococcus do not lyse in fresh water. We added a statement on the particularly sensitive behavior of H. walsbyi and N. pharaonic to saline conditions to the introduction. We also added the rationale above for the chosen dilutions to the methods.*

*Boring, J., Kushner, D. J., and Gibbons, N. E.: Specificity of the salt requirement of halobacterium cutirubrum, Canadian Journal of Microbiology, 9, 143-154, 10.1139/m63-020, 1963.*

*Leuko, S., Legat, A., Fendrihan, S., and Stan-Lotter, H.: Evaluation of the LIVE/DEAD BacLight Kit for Detection of Extremophilic Archaea and Visualization of Microorganisms in Environmental Hypersaline Samples, Appl Environ Microbiol, 70, 6884, 10.1128/AEM.70.11.6884-6886.2004, 2004.*

2. The authors argue correctly that, once airborne, halophiles would be exposed to a dilute environment. However, the salinity of the selected dilution is not representative of cloud droplets. Moreover, why intact cells that did show ice nucleation activity where not further diluted to cause cell lysis, and conversely, cells that were lysed were not tested at lower dilutions to keep cells intact is unclear.

*Reviewer 3 is exactly right, that once in a cloud droplet the salinity would be greatly lowered. Also, our experience of marine INPs always shows an increase in INA when diluted in DI water (above the 2 °C due to freezing point depression), as shown at right. The salinity chosen for the diluted culture tests was a reasonable compromise, typically from 0.5 – 1.5% (PBS is around 1% salt, and is not considered a particularly saline solution), to minimize freezing point depression while enabling expression if INA by the cells. As we stated in the introduction, H. morrhuae does not lyse in fresh water. And we did additional dilutions for H. morrhuae (Figure 2). See additional details in response to comment 1. We did omit the statement*

[Figure]

*about how the lowered salinities would be relevant for cloud droplets in sections 2.2, 3.1, and 4.*

3. The authors reported that diluting the media reduced survivability of halophiles and, in order to account for that, they determined the number of intact cells by microscopy. Yet, cell numbers reported in Table 1 are simply derived by multiplying cell numbers by the dilution factor (Table 2),

and thus, the study does not account for any losses caused by the dilutions. Undiluted cell suspensions should have been fixed to avoid cell lysis and counted. How can the authors account for a combined effect of intact cells and lysed cell material in these assays?

*We reported that diluting the media lysed all H. walsbyi and N. pharaonis. This was confirmed using microscopy. We also reported that diluting did not cause any significant losses to H. morrhuae or H. sulfurifontis. This too was confirmed using microscopy.*

4. The purpose of 30% H2O2 treatment is to determine the contribution of abiotic factors. However, the authors question the efficiency of the digestion protocol. If these 'digests' are a mixture of biotic and abiotic compounds, then there is no need for including the data.

*We disagree. The fact that not all organic INPs were effectively wiped out for H. morrhuae is interesting as it can indicate that there may be some contribution from inter-cellular materials that the peroxide is not able to reach in the more concentrated dilutions. Note that the only "abiotic compounds" would be from the media, since we know these cultures do not contain mineral dust. We already show the treatments on the media control for 1:15 H. morrhuae, which demonstrated there were no decreases in ice nucleation proficiency from the media (Figure 2b). Thus, it must be something biotic given the cultures only contained the media and the cells.*

*Based on the comments from Reviewer 1, we modified the end of this paragraph to expand on this point a bit more. We simply used a ratio of peroxide to sample suspension that has been demonstrated to work based on previous IS studies involving dilutions (e.g., Barry et al., 2021; Creamean et al., 2020; Suski et al., 2018). These studies have demonstrated that this ratio of peroxide has been sufficient to essentially knock out all organic INPs. Perhaps the archaea organic INPs are unique and some may be embedded within the cells, making it difficult for the peroxide to reach and thus degrade. We hope that any future investigations of archaeal INPs would include a more rigorous peroxide treatment regimen, akin to that of the recent Bogler et al. (2020) study on lignin biopolymer. We added a statement on our recommendation after this sentence in the revision and included a statement with references on the basis for choosing the amount of peroxide used.*

5. There is no statistical evidence presented that any of the organisms/treatments/controls were different albeit 24 replicates.

*This is incorrect. A difference in fraction frozen of ≥ 0.25 results in statistically significant differences according to Fisher's Exact test (p < 0.0479). That is more than clear for all the spectra in Figure 1. See response below for "Line 174-" for explanation regarding statistical significance of the differences between the treated and unamended spectra.*

*Again, we did not conduct comparisons between different dilutions or treatments for the four species. It is clear from Figure 1 and the corresponding text, when we do show the species under the same testing conditions, that there is a notable difference in the two intact versus two lysed species, especially when comparing to the DI water and media controls for each.*

As intact cells for H. walsbyi and N. pharaonic were not investigated and lysed cells did not show ice nucleation activity, they do not contribute to the study and should be removed.

*We disagree. No results are still results and warrant reporting in the literature; removal would simply be cherry picking results. See responses above regarding the inherent effects of dilutions and lysing for these two species.*

Methodology: L116: It is confusing and unclear why it is relevant the cells were first grown at the College of Charleston and then shipped to Colorado State University.

*We agree this is irrelevant and ended up removing the information on institutions where the cultures were grown and shipped.*

L116: Unclear why cells were first grown to mid-log phase but subsequently to somewhere during log-phase.

*They were not future grown after reaching mid-log phase. We now clarify that we tested the cultures as soon as possible after they reached this stage at the end of section 2.1: ". All cultures were grown at 37 °C and 100 rpm until mid-log phase (i.e., midway through the period of exponential cell growth). The purity and cell density were monitored optically at 1000× magnification throughout. Table 1 provides the cell concentrations and salinities of all four prepared cultures. Cultures were measured for their ice nucleation abilities within 1 – 2 days upon achieving log-phase growth (i.e., the period characterized by cell doubling)."*

L117: Please provide more detail no monitoring cell density. What microscope, cell counts, add reference on microscopy-cell abundance procedure. Also, the objective was most likely a 100 x with an extra 10x magnification within the eyepiece or camera. Unclear why cell density would be of relevance prior to shipping?

*Cell densities were obtained prior to shipping to ensure that the cultures were in mid-log phase. Also, to confirm that no appreciable growth had occurred between shipping and sample analysis.*

*The cells were counted using a Leica DM750 microscope (https://us.leica-camera.com/) at 1000x magnification with a 100x oil immersion objective lens. Cells were counted on a Petroff-Hausser 3900 counting chamber (http://hausserscientific.com/). This information is now included in section 2.1.*

L117: reference to Table 1: cells were shipped to a different university and additional experiment where performed at this university I assume. Cell numbers always remained constant during shipping, storage, and time passed until experimental setup?

*We monitored the cell numbers throughout, including once they arrived in Colorado immediately prior to the ice nucleation measurements, which we now provide more detail on in section 2.1. The numbers reported in Table 1 are from the most recent count.*

Table 1: Is salinity presented as gram NaCl? I am asking because e.g. DSMZ media 97 contains 250 g NaCl. Should this be 250 ppt and 25%.

*Thank you for bringing this to our attention. This culture of H. morrhuae was grown with only 150 g of NaCl. The alteration to media 97 is now included in the methods.*

L127: If the current study truly determined the lysis thresholds, why did the authors not in include dilutions for H. walsbyi and N. pharaonic. Cells were diluted 1:15 one time not serially. Dilution resulted in the lysis of two out of four selected organisms. Reason why not including lower dilutions that left cells intact as well as including dilutions for the other two organisms that would have resulted in cell lysis is unclear; particularly since the authors determined lysis thresholds.

*See response to comment 1.*

L128: Why were cells grown again if they already reached the desired cell density prior to shipping (L117)?

*The cells were not grown again. To prevent confusion, we deleted "active log-phase".*

Ll29: What was the reason for selecting two more dilutions for H. morrhuae but not the other three organisms? It is well known that cell density has an effect on ice nucleation.

*The goal was to test the most active of the four species, which we now clarify at the beginning of section 3.2: "Dilutions were only applied to H. morrhuae given its relatively high ice nucleating ability compared to the controls and the other haloarchaea tested."*

*On the contrary, according to our results in Figure 2, dilutions and thus cell density did not influence ice nucleation (i.e., the unamended spectra are roughly the same between the three dilutions.*

L130: add detail on microscopy.

*Done.*

L138: what was the coolant?

*Syltherm XLT. Now added to section 2.3.*

L142-148: (i) Heat treatment and 30% H2O2 amendments were intended to determine the effect lysed cell material and inorganic molecules on ice nucleation. If the authors think that the digestion was ineffective, they should have altered the protocol rather than hinting at the need for it. (ii) a general problem throughout the experimental design is the comparison of different cell 'material' i.e., intact, lysed, intact/digested, lysed/digested. It appears that the initial intact vs. lysed was an artifact and the authors went with it, but essentially, the study compares apples and oranges.

*(i) This is an inaccurate statement. Treatments were conducted on all four species, intact and lysed. We also did not think the digestion was ineffective, rather was only partially effective for only one species (H. morrhuae). See response to comment 4 above. (ii) See response to comment 1 above. The last statement in this comment is a crude speculation.*

Results and discussion: As the results and discussion section will largely change after removing a large portion of the dataset here are some general comments. When did controls freeze? H. morrhuae is atypical compared to all Archaea or the ones investigated?

*We are not removing a "large portion of the dataset" as contended in our responses above. Also, the controls are already provided in all the figures except for the 1:6 and 1:30 dilutions of the H. morrhuae media control, which we did not dilute further given the 1:15 dilution did not exhibit any decreases in freezing with the treatments. This makes sense given media 97 is 90% salts by weight, which is diluted down in 1000 mL of water and then further diluted to 1:15 here.*

*We do realize without reading the captions, it may not have been clear the media controls were already included, thus we revised to include more detailed legends in the figures.*

L167-172: Irrelevant as lyses cells are compared to intact cells. This is a study on INPs. No sure why it is 'interesting or relevant' to discuss cell lyses. As mentioned before low dilutions should have been used to not lyse cells.

*We disagree. See response to comment 1 above.*

L176-175: H2O2 treatment should provide information in the abiotic fraction not organic.

*The residual remaining is the abiotic fraction, yes, but the INPs removed are the organic INPs. Peroxide treatments (in addition to heat) remove stable organics and afford the contribution from inorganic materials or biological materials resistant to these treatments (Barry et al., 2021; Creamean et al., 2020; Hill et al., 2016; McCluskey et al., 2018; Perkins et al., 2020a; Suski et al., 2018; Tobo et al., 2014).*

L174-: Are any of these reported changes in freezing temperatures statistically different from intact cells?

*It is not clear what the reviewer is referring to here—H. morrhuae dilutions are all intact cells since it does not lyse in fresh water. For the differences in unamended to heat to peroxide, most of these differences are statistically significant according to Fisher's Exact test except for some of the heat treatment data for the 1:15 dilution (i.e., when approaching fraction frozen of 0 or 1 where there is less of a difference in temperature). A decrease in fraction frozen of ≥ 0.25 results in statistically significant differences (p < 0.0479). Certainly, all the peroxide spectra have a much higher decrease from the unamended spectra and thus are statistically significant differences. The statistical significance for each of the treated compared to unamended spectra is not discussed in section 3.2.*

L183-186: Discussion on ineffective H2O2 treatment. Effectively separating or removing specific fractions when investigating ice nucleation properties is essential. This section does not strengthen the manuscript.

*We disagree. See response to comment 4 above.*

L192-193: Why is N. pharaonis now only partially lysed contradicting previous statements?

*Thank you for bringing this to our attention, this was an error and in reference to a different dilution threshold. The text has been corrected.*

Conclusion: L209-213: seems more suited for an introduction.

*We do have this information in the introduction. It is reiterated here for emphasis on possible broader implications.*

Figure 1: Please remove the shading. The reader gets the impression that the area under the curve is of importance. Why on a log scale?

*Done. We also changed to linear scale.*

Figure 2: Why are there no controls for the dilutions 1:6 and 1:30? As shown in Table 2 the salinity ranges from 2.7%-0.5% in these samples.

*See response to the first general results and discussion comment above.*

Other edits: Check for missing punctuations

*Done.*

L10: delete 'from microorganisms'

*Done.*

L15-16: change 'of a subset of archaeal cells from Haloarchaea' to 'selected genera of the class Haloarchaea'

*Done.*

L16-18. Reason for comparing intact cells two lysed cells from different genera is unclear.

*See response to comment 1 above.*

L17: without comparing to the freezing temperature of an abiotic control, I would not consider -18C warm. Please rephrase.

*As indicated above, the controls were included. However, given –18 °C is not warm on a broader INP scale, we changed this to "at temperatures up to –18 °C".*

L18: What are warm temperature INPs? Please rephrase.

*We changed "warm temperature" to "immersion".*

L23: 'necessary to improve'. These are extremely strong words. How about 'intriguing'

*We instead rephrased to, "Thus, it is important to assess their ability to serve as INPs as it may lead to an improved understanding of biological impacts on clouds."*

L40: delete 'approximately'

*Done.*

L54: replace 'however' with 'further'

*Done.*

L57-61: please split this running sentence.

*Done. Changed to, "Even though many laboratory and field-based investigations have alluded to the importance of biologically-derived INPs, the relatively limited available observational data have caused models to produce equivocal results regarding the global significance of biological ice nucleation in cloud and precipitation formation (Burrows et al., 2013; Hoose et al., 2010b; Hummel et al., 2018; Phillips et al., 2009; Sesartic et al., 2012; Twohy et al., 2016; Vergara-Temprado et al., 2017). This modelling issue is further complicated by a very limited understanding and representation of secondary ice formation processes and their links to biologically-derived INPs in clouds."*

L64: Do all minerals except for feldspars function as INPs? I suggest deleting 'aside some feldspars'

*Good question. We have omitted "aside some feldspars".*

L71: delete 'relatively'

*Done.*

L82-83: Are there some bacterial cell that produce no peptidoglycan as the authors say 'nearly ubiquitous'?

*For an example, the mycoplasma and some other pathogens lack a cell wall entirely.*

L93: 'possess'

*Fixed.*

L96: replace 'it is' with 'its'

*Done.*

L97: For the other three genera cell characteristics were briefly described as a justification to include them in this study. Why not for Natronomonas?

*To our knowledge, there is no information in the literature on the cell wall structure for Natronomonas, which is why the description is limited.*

L100: change to 'they are relatively easy to culture compared to other archaeal lineages'

*Done.*

L105: What do the authors mean by 'halophiles and hypersaline'

*Typo. We removed "and hypersaline".*

L106-109: I recommend removing this paragraph. This paragraph is trying to oversell the importance of this research.

*This is the opinion of one reviewer, which we disagree with. Demonstrating broader significance is an important part of any manuscript.*

L116: delete '(i.e., midway through the period of exponential cell growth)'

*Done.*

L118: delete '(i.e., the period characterized by cell doubling)'

*Done.*

L141: delete 'for all species during each experiment.'

*We deleted "during each experiment" since we did run controls for all aside from the 1:6 and 1:30 dilutions for the reasons provided above.*

[revised manuscript text omitted]

---

## Author Response (AR1)

*Dr. Akob, we would like to thank you for your great suggestions for improvement on our manuscript, which we have revised further based on your feedback. Please find our responses to your comments and changes to the manuscript below in blue text. A track changes version is also included.*

**Specific Comments:**

1. Title: reviewer #2 suggests amending the title by replacing "Haloarchaea" for Archaea. I think this would be a valuable change and would highlight the focus of your study.

*Great suggestion, done.*

2. L. 15-16: consider revising to "nucleation activity of 4 species in the class Haloarchaea".

*Done.*

3. L. 10: change to "plants"

*We were not sure what you wanted changed to "plants". We ended up removing "plants" and included some additional microorganisms that we have in Figure 4 for consistency.*

4. L. 57, 72 and elsewhere: when referring to Bacteria (and Archaea and Eukaryotes) as a domain please capitalize. There are a number of instances where this change is needed for accuracy. Also, if Haloarchaea is a formal taxonomic group name it should always be capitalized.

*Done, for when referring to their domains / classes.*

5. L. 95: change to "these four"

*Done.*

6. Tables 1: It's not clear from the table and the methods text at which time the cell counts were made. In response to Reviewer 2, text was omitted on shipping the cultures from SC to CO. I disagree with this decision. The reviewer is pointing out a key methodological consideration -- even though you shipped the samples on ice it is important to know if the cultures were still in log-phase growth. It is super important to know what the cell abundance was at the time the experiments were started and how cell abundance changed during shipping. Even though you shipped the samples on ice there could be cell growth or death. Line 130 talks about the cultures being measured for ice nucleating ability within a few days of reaching log-phase but was this before or after shipping? Were the cultures still in log phase? More detail is needed.

*Thank you for clarifying how we should handle the issue of transport. Samples were monitored for growth at the College of Charleston. They were shipped overnight on ice to Colorado State where they were stored for up to 48 hours at 4°C. They were then checked for growth again and no change in cell density was observed. We have updated the text to better communicate our efforts at maintaining consistent culture environments. The text now reads: "Cells were counted and monitored for growth until mid-log phase at which point, they were shipped on ice overnight to Colorado and stored for up to 48 hours at 4 °C. In the event that cell densities were too high to achieve an accurate count, cultures were diluted to a countable level. Cultures were checked a final time for cell density immediately prior to ice nucleation assays to ensure that no appreciable growth had occurred during transport and storage. Table 1 provides the cell concentrations and salinities of all four prepared cultures immediately prior to ice nucleation assays."*

7. Table 1: why didn't you dilute the cultures in media to count them? That would have prevented cell lysis and give you an accurate cell count. Or you could have fixed the cells before diluting for cell

counts as suggested by Reviewer #3. Clearly you can't go back and redo this but consider this for future experiments.

*Cells were diluted to a factor of 1:6 for counting (except for H. morrhuae which could readily take a dilution in excess of 1:15). Since the microscopy check upon full dilution was primarily to assess whether cells remained intact, at the time we didn't think it was necessary to fix the cells or further dilute with a saline solution or media to protect against lysis. In retrospect, a second check on cell densities would have been helpful and we will certainly consider this for future experiments.*

8. Tables 1 and 2: I wonder if you really need 2 tables? The culture medium is already stated in the methods so could be omitted. You could combine the information into a single table and have columns of initial and diluted cell concentrations and salinities. I also suggest putting information in order of action, e.g., you diluted before knowing the cells were intact so switch the column order; otherwise it makes it sound like you diluted lysed cells.

*Good idea, done.*

9. L. 105-112: I think this paragraph is of value and should be retained. The authors do a nice job here of identifying how the lab cultures can be environmentally relevant.

*Thank you!*

10. L. 137-142: a lot of the reviewers' comments were around the different treatments and use of lysed vs. intact cells. The text here is a great justification on why the treatments make sense and that the different cultures behaving differently has environmental relevance. But, this message really isn't stated in the introduction. It would improve the paper if the statement about testing cell lysis having environmental relevance was included somewhere in the end of the intro.

*Done. We have added the following sentence towards the end of the introduction when first mentioning cell lysis: "Assessing a variety of cells that lyse or remain intact is relevant for ice nucleation because cell fragments of other microorganisms have been shown serve as INPs and (Anderson and Ashworth, 1986; Du et al., 2017; O'Sullivan et al., 2015; Šantl-Temkiv et al., 2015) and archaea might lyse naturally once exposed to atmospheric water vapor in the aerosol phase."*

11. L. 144-148: the added text is very helpful.

*Glad to hear, thank you.*

12. L. 142: I'm not sure if I get what is meant by "threshold" and "serially dilution" here – Table 2 doesn't show data for serial dilutions, just a few select samples. If the cultures were serially diluted that would indicate to me that there is a whole series of dilutions that were assessed for cell survival and the highest dilution that didn't show lysis was the threshold. Are you really just presenting the final dilution selected based on those factors? Or did you only select these dilutions based on tests with saline solution controls? Please clarify.

*Thank you for bringing this to our attention. For clarification, we removed "serially" since these were indeed the only dilutions we created and tested. We also changed "Cell lysis thresholds were determined…" to simply "Cell lysis was determined…".*

13. L. 152: please define the abbreviation CSU

*Done (Colorado State University).*

14. Section 3.2: I found the start of this section to be overly focused on the dilutions taking away from the study aim. I suggest revising to start with the main finding and not the methodology. Also, the use

of "dilution" seems a bit mis-leading since the focus is really on the cell density and not the salinity of dilution of the sample.

*We moved some of the text at the beginning of section 3.2 to the methods. We now discuss the findings and try to avoid using dilution as much as possible by clarifying that increasing dilution = decreasing cell density. When intercomparing the dilutions, we changed "dilution" to "sample" (e.g., "1:6 dilution" to "1:6 sample").*

15. L. 216 and elsewhere: the original paper did not have statistics presented. Adding the fisher's exact test was really helpful for showing the differences in the treatments. Consider adding a supplemental table of the statistics results to compliment the narrative.

*We have now compiled a table for statistics for the treatments and will upload as a supplementary file and referenced to in the text (Table S1).*

16. L. 266: change to "range tested" as it would be a stronger statement

*Good suggestion, done.*

17. L. 269: cite the references on H. morrhuae cells not lysing here.

*Done.*

18. L. 270-272: the statement on the DUF3494 protein domain in the Archaea does not belong here. The reference is not from any of your authors, and you don't show any data. If you want to include this observation the results and methods need to be included.

*Vance et al. (2019) indicated that archaea can contain this protein, but perhaps this is too vague to make a link to Haloarchaea. We also did not conduct any such observations, so we have removed the reference to DUF3494.*

19. L. 294: based on this acknowledgement it sounds like the organisms were cultured in Colorado and not just in SC. If they were preserved does that indicate cells were fixed before counting? Clarification would be really helpful per the comments above about the methods for culturing and shipping.

*This reflects their efforts on a previous iteration of the experiment that is not discussed in the manuscript. This acknowledgement has been removed.*

20. Table 2: correct the spelling of pharaonis in the footnotes

*Fixed.*

21. Figure 3: the revised, track changes ver

sion of the paper shows 2 graphs with different data presented. According to a response to Reviewer #1 the data in Figure 3 were originally mislabeled but are now fixed and the text is correct. However, please double check this for accuracy during revision.

*Reviewer 1 was correct in that we did mix up the symbols and colors originally, which we fixed for the revision. We have gone back to the raw data files and confirmed that the data shown in the revision are correct.*